# Hierarchical architecture of dopaminergic circuits enables second-order conditioning in *Drosophila*

Daichi Yamada[1], Daniel Bushey[2], Feng Li[2], Karen L Hibbard[2], Megan Sammons[2], Jan Funke[2], Ashok Litwin-Kumar[3], Toshihide Hige[1,4,5]*, Yoshinori Aso[2]*

[1]Department of Biology, University of North Carolina at Chapel Hill, Chapel Hill, United States; [2]Janelia Research Campus, Howard Hughes Medical Institute, Ashburn, United States; [3]Department of Neuroscience, Columbia University, New York, United States; [4]Department of Cell Biology and Physiology, University of North Carolina at Chapel Hill, Chapel Hill, United States; [5]Integrative Program for Biological and Genome Sciences, University of North Carolina at Chapel Hill, Chapel Hill, United States

**\*For correspondence:**
hige@email.unc.edu (TH);
asoy@janelia.hhmi.org (YA)

**Competing interest:** The authors declare that no competing interests exist.

**Abstract** Dopaminergic neurons with distinct projection patterns and physiological properties compose memory subsystems in a brain. However, it is poorly understood whether or how they interact during complex learning. Here, we identify a feedforward circuit formed between dopamine subsystems and show that it is essential for second-order conditioning, an ethologically important form of higher-order associative learning. The *Drosophila* mushroom body comprises a series of dopaminergic compartments, each of which exhibits distinct memory dynamics. We find that a slow and stable memory compartment can serve as an effective 'teacher' by instructing other faster and transient memory compartments via a single key interneuron, which we identify by connectome analysis and neurotransmitter prediction. This excitatory interneuron acquires enhanced response to reward-predicting odor after first-order conditioning and, upon activation, evokes dopamine release in the 'student' compartments. These hierarchical connections between dopamine subsystems explain distinct properties of first- and second-order memory long known by behavioral psychologists.

## Editor's evaluation

Second-order conditioning is a higher form of learning where a previously conditioned stimulus is used to condition the perception of another stimulus. The authors have used the fly to identify a neural circuit in the insect mushroom body underpinning this fundamental ability of higher animals. This important work elegantly combines neural circuit mapping, electrophysiology, and modelling to put forward a compelling, mechanistic model for this highly conserved form of learning.

## Introduction

Knowledge about order and regularities in environments is crucial for animal survival. Although direct temporal correlation between stimuli and rewards is a primary drive for associative learning, animals are also capable of learning indirect relations between stimuli and rewards in many real-life situations. For example, bumble bees, who have prior foraging experience with other bees, can learn to visit a flower of a particular color without tasting nectar just by watching other bees sitting on flowers of that color (*Avarguès-Weber and Chittka, 2014*; *Worden and Papaj, 2005*). In the case of humans, some

TV commercials can be considered as conditioning of consumers to associate items with the positive valence that has been already associated with popular cartoon characters. In both cases, learning depends on the valence of stimuli (i.e. sight of other bees or cartoon characters) that is acquired through prior experience. Although such higher-order associative learning is widely observed across species and ethologically important, its circuit mechanisms are poorly understood compared to those of simpler forms of associative learning.

Second-order conditioning is a major form of higher-order associative learning. In this learning paradigm, an initially neutral stimulus is paired with reward or punishment; that stimulus, which is now predictive of reward/punishment, then serves as an effective reinforcer when learning about a new stimulus. Since Pavlov's classic experiment with dogs (*Pavlov and Gantt, 1927*), second-order conditioning has been demonstrated in various vertebrate and invertebrate models (*Bitterman et al., 1983*; *Brembs and Heisenberg, 2001*; *Hawkins et al., 1998*; *Holland and Rescorla, 1975*; *Mizunami et al., 2009*; *Rizley and Rescorla, 1972*; *Sisk, 1976*; *Tabone and de Belle, 2011*; *Takeda, 1961*). Furthermore, second-order conditioning is thought to extend the applicability of Pavlovian conditioning as an account of behaviors including observational learning (*Avarguès-Weber and Chittka, 2014*; *Worden and Papaj, 2005*). Additionally, second-order conditioning has also served as a historically important tool for behavioral psychologists to study associative learning by giving them ample options to use virtually any stimulus as a reinforcer (*Rescorla, 1980*).

One prominent feature that characterizes second-order memory is its transiency, as originally noted by Pavlov and confirmed by other studies using various animal models (*Herendeen and Anderson, 1968*; *Stout et al., 2004*; *Yin et al., 1994*). That is, the effectiveness of second-order conditioning usually reaches an asymptote after a small number of trials and begins to decline with further training (*Gewirtz and Davis, 2000*; *Pavlov and Gantt, 1927*). This decline may be related to the fact that reward is constantly omitted during second-order conditioning. Another important feature of second-order conditioning recognized by behavioral psychologists is that it does not form a tight association between the stimulus and the specific response elicited by the reinforcer, which is typically observed in first-order conditioning (*Gewirtz and Davis, 2000*; *Pavlov and Gantt, 1927*). In other words, second-order learning seems to be based on general valence, rather than specific features, of reinforcers. These differences between first- and second-order memories raise important mechanistic questions: What is the circuit origin of those different memory features? Are they different because those two memories are stored in separate circuits that support distinct types of memories? If so, how do the two circuits interact when one memory instructs the other? Answering these questions requires precise mapping of second-order memory circuits.

In rodents, basolateral amygdala and dopaminergic neurons (DANs) play critical roles in second-order learning (*Gewirtz and Davis, 1997*; *Maes et al., 2020*). After first-order association, DANs in the ventral tegmental area acquire enhanced responses at the onset of the cue that predicts upcoming reward after conditioning (*Schultz, 1998*). A recent study used optogenetic silencing to demonstrate that such cue-evoked dopamine transients are essential for second-order conditioning (*Maes et al., 2020*). Whereas DANs consist of functionally diverse populations of neurons, each of which contributing to distinct types of learning (*Roeper, 2013*; *Watabe-Uchida and Uchida, 2018*), how these different DAN subtypes interact during second-order conditioning is completely unstudied.

The *Drosophila* mushroom body (MB), a dopamine-rich center for associative learning in insect brains, provides a tractable system to study the interaction between heterogeneous dopamine subsystems. *Drosophila* can perform second-order learning using olfactory or visual cues with punishment (*Brembs and Heisenberg, 2001*; *Tabone and de Belle, 2011*), although the underlying circuit mechanisms have not been examined. Decades of studies have revealed the anatomical and functional architecture of the MB circuit (*Figure 1A*). Along the parallel axonal fibers of Kenyon cells (KCs), DANs and MB output neurons (MBONs) form 16 matched compartments (*Aso et al., 2014*; *Li et al., 2020*; *Tanaka et al., 2008*), which serve as units of associative learning. Reward and punishment activate distinct subsets of 20 types of DANs (*Berry et al., 2015*; *Burke et al., 2012*; *Kirkhart and Scott, 2015*; *Lewis et al., 2015*; *Lin et al., 2014*; *Liu et al., 2012*; *Riemensperger et al., 2005*; *Siju et al., 2020*). Individual DANs write and update memories in each compartment with cell-type-specific dynamics by modulating synaptic connection between KCs and MBONs (*Aso et al., 2019*; *Aso et al., 2012*; *Aso and Rubin, 2016*; *Hige et al., 2015*; *Huetteroth et al., 2015*; *Owald et al., 2015*; *Perisse et al., 2016*; *Vrontou et al., 2021*; *Yamagata et al., 2015*). Outside the MB, MBON

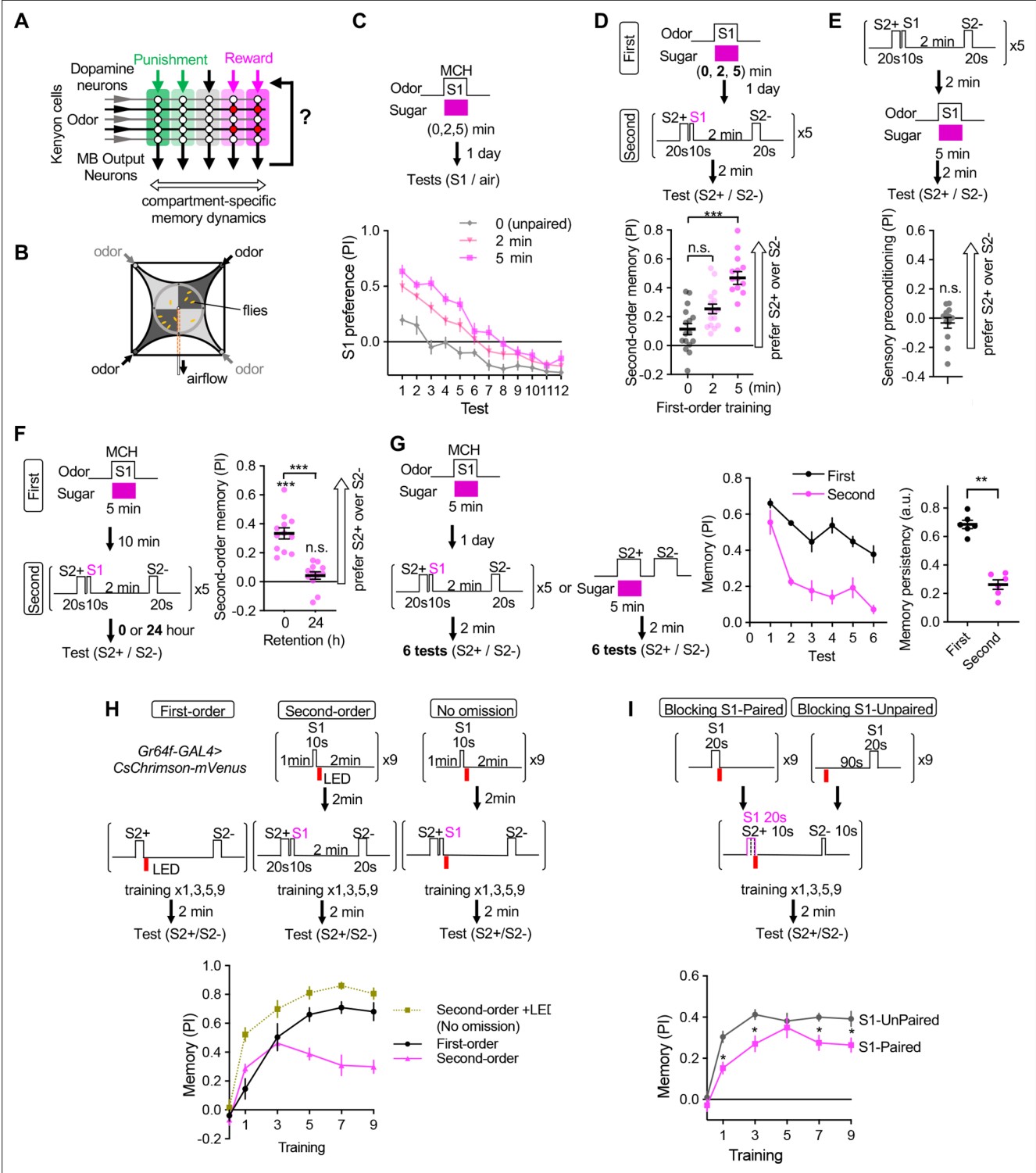

**Figure 1.** Appetitive olfactory second-order conditioning in *Drosophila*. (**A**) A simplified diagram of the mushroom body circuit. Identity of odors are encoded by patterns of activity in ~2000 Kenyon cells. Contingent activity of Kenyon cells and dopamine release leads to plasticity of excitatory synapses from Kenyon cells to MB output neurons with compartment-specific dynamics. (**B**) A diagram of the four-armed olfactory arena. Flies were confined in the 9 cm diameter circular area above the LED board. For odor-sugar conditioning, flies were first trained in a tube by pairing an odor with dried sugar paper, and then introduced to the olfactory arena. Performance index was calculated by counting the number of flies in each quadrant (see Methods). (**C**) Dynamics of MCH preference after various 2 or 5 min of first-order conditioning with sugar. Flies were trained after 40–48 hr of starvation and memories were tested 20–24 hr later without feeding in between by examining preference to MCH over air for 12 times. Unpaired group

*Figure 1 continued on next page*

*Figure 1 continued*

received 5 min of sugar 2 min prior to 5 min exposure to MCH. Mean performance index of the first 5 tests after 5 min training was higher than that of 2 min. p<0.01; unpaired t-test; N=10–12. (**D**) Second-order memory performance by wild type flies. n.s., not significant (p=0.152); ***, p<0.0001; Dunn's multiple comparison tests following Kruskal-Wallis test; N=14–16. Means and SEMs are displayed with individual data points. (**E**) The odor preference following the sensory preconditioning protocol, in which the order of the first and second-order conditioning was swapped. n.s., not significantly different from the chance level; Wilcoxon signed-rank test; N=12. (**F**) Retention of second-order memory. After 24 hr, the second-order memory decayed to the chance level. ***, p<0.001; Wilcoxon signed-rank test or Mann-Whiteney test; N=12. (**G**) Odor preference between two S2 odors after the second-order or first-order conditioning was measured for six times by alternative position of two odorants with 2 min intervals. Memory persistency, a mean of PIs for 3rd-6th tests divided by PI of 1st test, was significantly smaller for second-order memory. **; p<0.0022; Mann-Whitney test; N=6. Means and SEMs are displayed with individual data points. (**H**) Learning curves by first-order, second-order, or second-order without omission of optogenetic reward. Flies expressing CsChrimson in sugar sensory neurons with Gr64f-GAL4 were trained by pairing S2+ odor with activation of LED (First) or S1 odor that was previously paired with LED (Second). In the no omission protocol, sugar sensory neurons were activated immediately after S1 by repeating 1 s red LED illumination with 1 s intervals for three times. Preference between S2+ and S2− odors was tested after 1st, 3rd, 5th, 7th, and 9th training sessions. After 9th training, memory by second-order protocol was lower than other protocols and its peak at 3rd training (p<0.05); Dunn's multiple comparison tests following Kruskal-Wallis test; N=8. (**I**) Learning of S2 odors was compromised when S1 odor paired with Gr64f>CsChrimson precedes S2+ odor. *, p<0.05 by Dunn's tests following Kruskal-Wallis test; N=12.

The online version of this article includes the following source data and figure supplement(s) for figure 1:

**Source data 1.** Numerical data for *Figure 1*.

**Figure supplement 1.** Dynamics of odor preference.

**Figure supplement 1—source data 1.** Numerical data for *Figure 1*.

axons project to regions where DAN dendrites arborize; this provides an anatomical pathway for feedback of memory-based information onto DANs, a potential substrate for higher-order conditioning. Indeed, early studies showed that DANs in the MB dynamically change odor responses after olfactory conditioning (*Riemensperger et al., 2005*). Furthermore, the recently completed EM connectome (*Scheffer et al., 2020*) revealed the full wiring diagram of the MB, including intricate connections from MBONs to the DANs. In both larval and adult *Drosophila*, large fractions of synaptic inputs to the MB's DANs originate from the MB itself (*Eschbach et al., 2020*; *Li et al., 2020*). Thus, it is plausible that induction of synaptic plasticity in one compartment, in turn, affects how a learned stimulus activates DANs and becomes a secondary reinforcer. However, understanding the flow of information across compartments that underlies second-order conditioning is a challenging task, given that thousands of neurons are connected with DANs and MBONs.

Here, by exploiting connectomic data, we identify a key circuit that underlies second-order conditioning. We first establish a protocol for robust olfactory second-order conditioning with sugar reward. In contrast to stable odor-sugar first-order memory, second-order memory decayed within a day and was highly susceptible to extinction. We next show that memory in α1, the compartment responsible for long-lasting appetitive memory (*Ichinose et al., 2015*; *Yamagata et al., 2015*), is most potent to promote second-order memory. The second-order memory instructed by α1 was transient during the training phase and extinction trials. Subsequent EM connectome and functional analysis identify a prominent cholinergic interneuron SMP108 that (1) forms an excitatory pathway from MBON-α1 to DANs in other compartments, (2) acquires an enhanced response to the reward-predicting odor, (3) can promote release of dopamine in multiple compartments, (4) is required for second-order conditioning, and (5) induces memory with fast and transient dynamics. Our study reveals in unprecedented detail circuit mechanisms of second-order conditioning. These mechanisms can explain the different properties of first- and second-order memories. They also provide a concrete example of how hierarchical interaction between dopamine subsystems contributes to a complex form of learning.

## Results
### Olfactory second-order conditioning following the odor-sugar association

As a prerequisite for mapping the underlying neuronal circuits and detailed characterization of memory properties, we established a robust protocol for appetitive second-order conditioning using a circular olfactory arena (*Figure 1B* and *Figure 1—figure supplement 1*; see Methods for our rationale for the selection of odors and other parameters). Flies were first trained to associate stimulus one (S1) odor

with sugar and consolidated that memory for 1 day (*Figure 1C*). During second-order conditioning, 20 seconds of one S2 odor (S2+) was immediately followed by 10 s of the S1 odor, whereas another S2 odor (S2−) was presented alone. After five training sessions, flies increased their preference to the S2+ odor over the S2− odor when first-order conditioning was long enough (i.e. 5 min; *Figure 1D*). This preference for the S2+ odor was not due to sensory preconditioning, another form of higher-order conditioning in which S2-S1 pairing was done *before* pairing S1-sugar (*Figure 1E*), although unimodal sensory preconditioning has been reported in aversive olfactory learning in *Drosophila* (*Martinez-Cervantes et al., 2022*).

First-order memory and its derived second-order memory exhibited marked differences in dynamics of formation and update. Second-order memory after odor-sugar conditioning did not last for one day and was susceptible to extinction (*Figure 1F and G*). With optogenetic stimulation of sugar sensory neurons, the first-order memory steadily increased during nine training sessions, whereas second-order memory peaked at the third training and declined subsequently (*Figure 1H*). This transiency of learning was not observed when activation of sugar sensory neurons was not omitted during second-order conditioning (*Figure 1H*). Learning of association between S2+ odor and activation of sugar sensory neurons was compromised when S2+ is preceded by S1 which predicts the occurrence of reward (*Figure 1I*). These results indicate that the transient and unstable nature of second-order memory observed across animal phyla also applies to *Drosophila*, and the temporal order of the stimuli is crucial for second-order conditioning as in first-order conditioning.

## Identification of MB compartments that instruct second-order conditioning

To identify the circuit elements that might be particularly important for second-order conditioning, we examined whether first-order memory in certain MB compartments is more potent for instructing second-order conditioning than others. For this purpose, we substituted sugar with optogenetic activation of DANs to induce memory in a defined set of compartments (*Figure 2A*). Flies were first trained by pairing the S1 odor with optogenetic activation of specific DANs with CsChrimson.

(see below for measurement of dopamine release). Then, the compartment-specific memory of the S1 odor was tested for its power as a reinforcer in second-order conditioning. Among four sets of DAN cell types that can induce first-order appetitive memory (*Figure 2A*), two sets — PAM-α1 and a combination of PAM-γ5 and β′2a — could induce significant second-order memory compared to the genetic control (*Figure 2B*). Similar to first-order conditioning, stimulus timing was an important factor for successful second-order conditioning (i.e. S2+ must precede S1; *Figure 2C*). PAM-α1 is known to be essential for learning nutritional value and is required for long-term appetitive memory (*Yamagata et al., 2015*), whereas memory induced by combinatorial activation of PAM-γ5 and PAM-β′2a is short-lasting (*Aso and Rubin, 2016*). As expected from those different stabilities of the first-order memory, memory in PAM-α1 but not PAM-γ5/β′2a could instruct second-order conditioning one day after the first-order conditioning (*Figure 2B*). Consistent with the outcome of this optogenetic experiment, blocking of synaptic transmission from PAM-α1 DANs with Tetanus Toxin (TNT) light chain abolished both S1 preference and second-order memory when assayed one day after odor-sugar conditioning (*Figure 2D*). In contrast, blocking PAM cluster DANs in the γ4, γ5, β′2a with TNT impaired the second-order conditioning without affecting S1 preference (*Figure 2D*). The second-order memory derived from the first-order memory in the α1 compartment exhibited the transient learning curve (*Figure 2E–F*) and susceptibility to extinction, recapitulating observations after odor-sugar conditioning (*Figure 1F–H*). Thus, these results suggest α1 as the primary candidate compartment to store the first-order memory that instructs second-order conditioning. The first-order memory in the γ5/β′2 a compartments may have a supplemental contribution to second-order conditioning, especially shortly after the first-order conditioning.

## Memory in α1 can instruct secondary plasticity across compartments

Memories and plasticity induced in different MB compartments differ in their properties including retention, induction threshold and resistance to extinction (*Aso et al., 2012*; *Aso and Rubin, 2016*; *Hige et al., 2015*; *Huetteroth et al., 2015*; *Jacob and Waddell, 2020*; *Lin et al., 2014*; *Pai et al., 2013*; *Plaçais et al., 2013*; *Vrontou et al., 2021*; *Yamagata et al., 2015*). The markedly distinct memory dynamics between first- and second-order memories noted above prompted us to

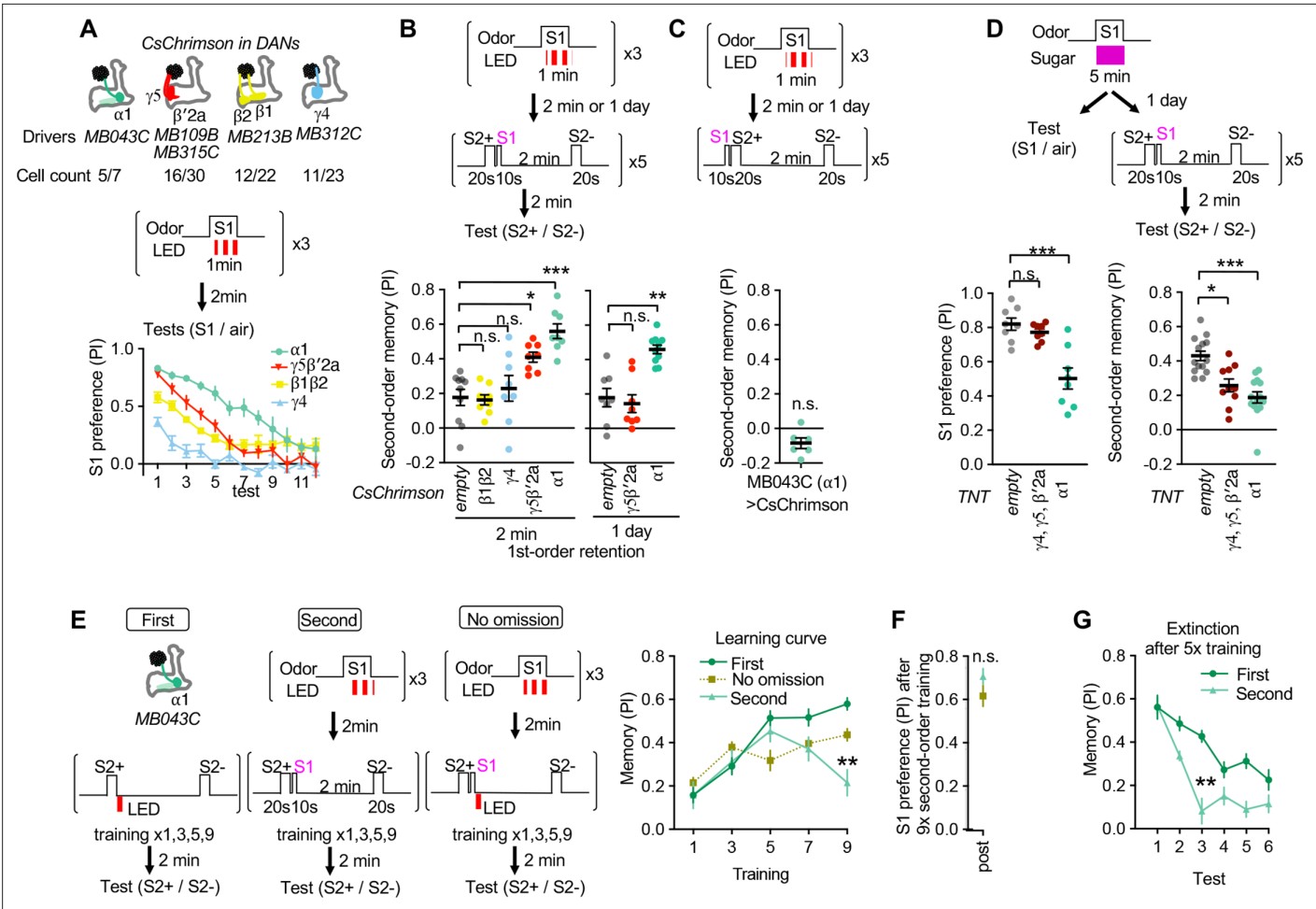

**Figure 2.** Identification of the teacher compartment(s). (**A**) Dynamics of S1 odor (MCH) preference after pairing 1 min of S1 odor with activation of different PAM-cluster DANs for three times. Numbers of CsChrimson-mVenus in each driver per hemisphere and total number of corresponding DAN cell types in EM hemibrain data are indicated. At 3rd-7th tests, MCH preference of MB043C>CsChrimson flies was higher than all other genotypes. p<0.05; Dunn's multiple comparison tests following Kruskal-Wallis test; N=6. (**B**) The second-order conditioning 2 min or 1 day after the first-order conditioning with optogenetic activation of various DAN types. Second-order memory was tested immediately after pairing S2+ odor with S1 odor (MCH) five times. n.s., not significant; *, p=0.0330; **, p=0.0046 ***, p<0.001; Dunn's multiple comparison tests following Kruskal-Wallis test; N=8–10. (**C**) The second-order memory immediately after backward second-order conditioning. Flies expressing CsChrimon-mVenus by MB043C split-GAL4 were trained with identical protocol as in B, except that the onset of S1 odor was shifted to the 10 second before the onset of the first S2 odor. n.s., not significant from zero; Wilcoxon signed-rank test; N=6. (**D**) Preference to the S1 odor (left) and second-order memory (right) by flies expressing TNT with empty, MB196B or MB043C split-GAL4. MB196B labels ~ 27 cells per hemisphere, including PAM-γ4, PAM-γ4<γ1γ2, γ5 and β'2a. *, p=0.0126; ***, p<0.001; Dunn's multiple comparison tests following Kruskal-Wallis test; N=8 for S1 preference; N=10–14 for second-order. (**F**) Learning curves by first-order, second-order, or second-order without omission of optogenetic reward. Flies expressing CsChrimson with MB043C split-GAL4 were trained by pairing S2+ odor directly with optogenetic activation of DANs (First) or S1 odor that was previously paired with DAN activation (Second). In the no omission protocol, DANs were activated immediately after S1 by repeating 1 s red LED illumination with 1 s intervals for three times. Preference between S2+ and S2− odors was tested after 1st, 3rd, 5th, 7th, and 9th training sessions. After 9th training, memory by second-order protocol was lower than other protocols and its peak at 5th training. **, p<0.01; Dunn's multiple comparison tests following Kruskal-Wallis test; N=8–10. (**G**) The preference for the S1 odor (MCH) after the 9th session of second-order conditioning as in F. n.s., not significant; Mann-Whitney test; N=8. (**H**) Comparison of memory decay after repetitive tests. Flies were trained five times with first or second-order conditioning protocol as in F but without tests. Immediately after the 5th training, preference between two S2 odors was measured repeatedly without training. At third test, second-order memory was significantly lower than first-order memory. **, p=0.0036; Dunn's multiple comparison tests following Kruskal-Wallis test; N=8.

The online version of this article includes the following source data for figure 2:

**Source data 1.** Numerical Data for *Figure 2*.

hypothesize that those memories are formed in different MB compartments. For aversive memory, transient inactivation of MBON-γ1pedc (a.k.a MB-MVP2), which mimics the effect of synaptic depression caused by aversive learning, can serve as reinforcement (*König et al., 2019*; *Ueoka et al., 2017*). Thus, if our hypothesis is correct, and if the α1 compartment indeed is potent for instructing second-order conditioning, then local induction of synaptic plasticity in α1 should drive secondary plasticity in other compartments during second-order conditioning. Since PAM-γ5 and β′2a can induce robust appetitive memory that is short-lasting and susceptible to extinction (*Figure 2A*; *Aso and Rubin, 2016*), we reasoned that second-order memory may involve compartments targeted by these DANs. To test this idea, we first generated a split-LexA driver to express ChrimsonR selectively in PAM-α1 (*Figure 3—figure supplement 1*). We then labeled either MBON-α1 or MBON-γ5β′2a by split-GAL4 lines to make whole-cell recordings from them (*Figure 3A* and *Figure 3—figure supplement 2A*). In MBON-α1, we found that pairing an odor and DAN activation leads to reduced spiking responses to that odor as in other MB compartments examined in previous studies (*Figure 3—figure supplement 2*; *Berry et al., 2018*; *Handler et al., 2019*; *Hige et al., 2015*; *Owald et al., 2015*; *Owald and Waddell, 2015*; *Séjourné et al., 2011*; *Vrontou et al., 2021*). MBON-γ5β′2a, on the other hand, did not elicit action potentials that are readily distinguishable from synaptic potentials in response to odor presentation or current injection (*Figure 3—figure supplement 3*). We therefore focused on subthreshold responses. After a single round of second-order conditioning, MBON-γ5β′2a showed reduced responses to the S2+ odor, while responses to S2− did not change even after five repetitions of conditioning (*Figure 3B and C*). Repeated presentation of S2 odors without S1 did not cause a reduction of odor responses (*Figure 3D and E*). These results indicate that the α1 compartment can instruct second-order conditioning in the γ5/β′2a and potentially other compartments.

## Candidate interneurons to mediate instruction signals for second-order conditioning

We next set out to identify the neuronal pathway responsible for the induction of second-order plasticity. MBON-α1 is the sole output pathway from the α1 compartment and is, like other reward memory compartment MBONs, glutamatergic. Glutamate functions as an inhibitory neurotransmitter with glutamate-gated-chloride channel (*Liu and Wilson, 2013*), although activity of glutamatergic MBONs can have a net excitatory effect on DANs via other receptors or indirect pathways (*Cohn et al., 2015*; *Ichinose et al., 2015*; *Karuppudurai et al., 2014*; *Zhao et al., 2018*). Upon induction of plasticity, MBON-α1's responses to learned odor will be depressed (*Figure 3—figure supplement 2*). Therefore, if glutamate is inhibitory, the downstream circuits of the MBON-α1 could gain an enhanced response to a learned odor as an outcome of reduced inhibition, which could feed an excitatory drive to DANs for second-order conditioning, provided that there are such connections. However, α1 appears to be an exceptionally isolated compartment. MBON-α1 is the only MBON that does not send direct output to DANs innervating other compartments; rather it only directly connects with the DANs that innervate the same compartment, PAM-α1 (*Figure 4—figure supplement 1A*; *Li et al., 2020*). Similarly, MBON-α1 shows very limited connections to DANs innervating other compartments that are mediated by a single interneuron (one-hop pathways; *Li et al., 2020*; *Figure 4—figure supplement 1B*). This led us to explore pathways with two interneurons between MBON-α1 and DANs (two-hop pathways).

To explore pathways with interneurons between MBON-α1 and DANs, we queried the hemibrain EM connectome database (*Li et al., 2020*; *Scheffer et al., 2020*). We then used a pre-trained machine learning algorithm to predict the most likely neurotransmitters used by the connected neurons (*Eckstein et al., 2020*). *Supplementary file 1* summarizes the full connection matrix, neurotransmitter predictions for the 396 major interneuron cell types with at least 100 total synapses with MBONs and DANs. In this way (see Materials and methods for detail), we identified prominent cholinergic two-hop pathways from MBON-α1 to multiple reward-DANs including PAM-γ5, γ4, β′2a, β′2m, β′2p that were mediated by the interneurons SMP353/354 and SMP108 (*Figure 4A*; *Figure 4—figure supplement 2*). The SMP353/354 are a subset of the UpWind Neurons (UpWiNs) that transform appetitive memory into directional turning to the upwind orientation (*Aso et al., 2022*). The SMP108 is an outstanding cell type in many features. Among all cholinergic neurons, SMP108 has the highest number of connections with reward DANs (*Figure 4—figure supplement 3*). SMP108 also synapses onto all three cholinergic interneurons (SMP177, LHPV5e1, LHPV10d1) in the second layer of the two-hop

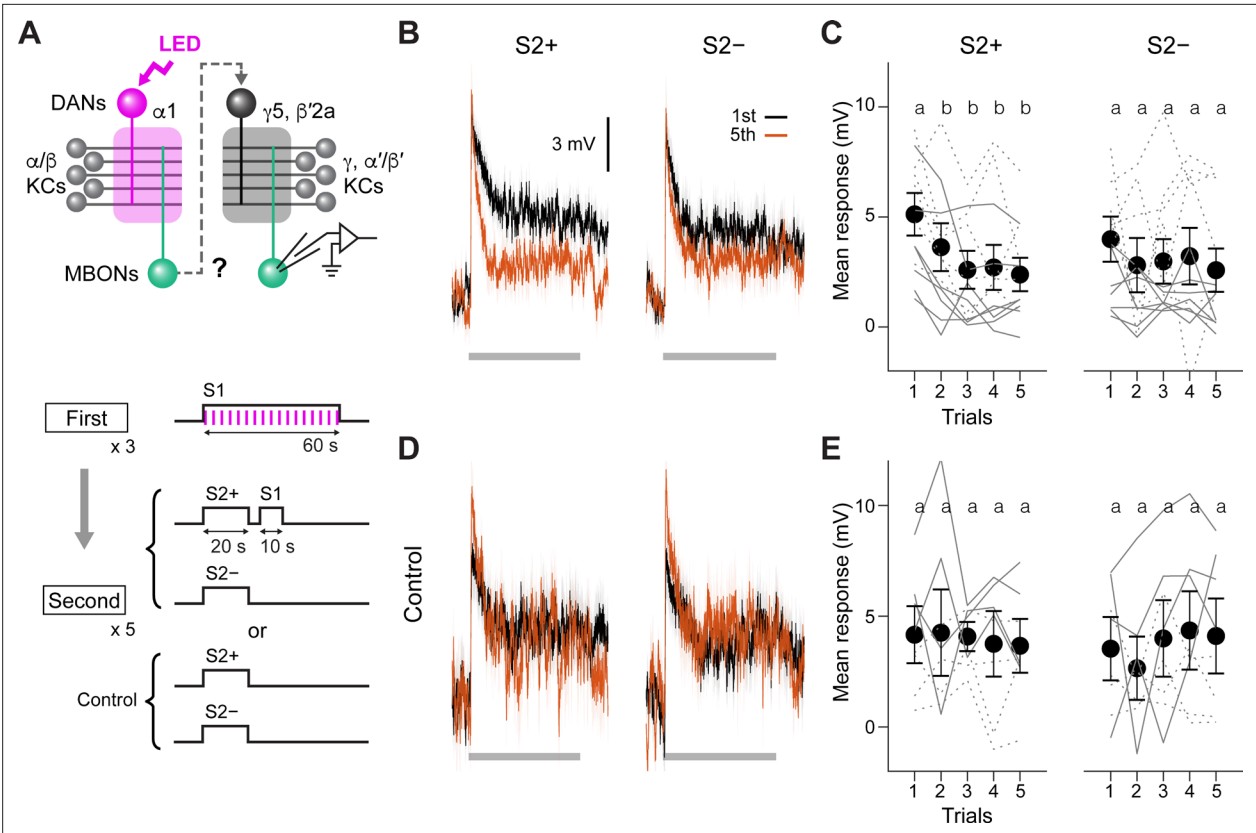

**Figure 3.** Second-order conditioning induces cross-compartmental plasticity. (**A**) Experimental design and protocol. ChrimsonR-mVenus was selectively expressed in PAM-α1 using MB043-split-LexA (*58E02-ZpLexADBD in JK22C; 32D11-p65ADZp in JK73A*; see **Figure 3—figure supplement 1** for expression pattern), and in vivo whole-cell recordings were made from MBON-γ5β'2a, which was labeled by mScarlet using a split-GAL4 driver *SS01308*. For the first-order conditioning, 1 min presentation of S1 (MCH) was paired with LED stimulation (1ms, 2 Hz, 120 times), which caused odor-specific suppression of responses in MBON-α1 (**Figure 3—figure supplement 2**). After repeating first-order conditioning three times with 2 min intervals, second-order conditioning was performed by presenting S2+ (either PA or EL) for 20 s, and then S1 for 10 s with 5 s delay. S2− was presented alone 2 min later. Second-order conditioning was repeated five times, and the responses to S2 were recorded. In control experiments, first-order conditioning was performed in the same manner, but the presentation of S1 was omitted during second-order conditioning. Reciprocal experiments were performed by swapping S2+ and S2− in separate flies. (**B**) Mean responses ( ± SEM in light colors) to S2+ and S2− in the first (black) and fifth trials (red) during second-order conditioning (n=14, including reciprocal experiments). Horizontal gray bars indicate 20 s odor presentation period. (**C**) Mean response magnitudes ( ± SEM) evoked by S2+ and S2−. The response magnitude was calculated by averaging the depolarization during the response window (0–20.6 s from odor onset). Each solid (PA used as S2+; n=7) and dashed line (EL as S2+; n=7) indicates data from a single fly. Responses to S2+ underwent depression after the first trial, while those to S2− did not change. Different letters indicate significant differences detected by Tukey's post hoc multiple comparisons test (p<0.05) following repeated-measures two-way ANOVA (p=0.003). There was no significant change in the peak amplitude (p=0.87). (**D, E**) Same as (**B**) and (**C**) except that the data are from control experiments (n=4 each with PA or EL used as S2+, respectively). Neither responses to S2+ nor S2− changed (p=0.28; repeated-measures two-way ANOVA). The peak response did not change either (p=0.22).

The online version of this article includes the following source data and figure supplement(s) for figure 3:

**Source data 1.** Numerical data for **Figure 3**.

**Figure supplement 1.** Expression patterns of MB043-split-LexA, MB319C and SS67221-split-GAL4.

**Figure supplement 2.** Optogenetic conditioning in α1 compartment induces depression in MBON-α1.

**Figure supplement 2—source data 1.** Numerical data for **Figure 3—figure supplement 2**.

**Figure supplement 3.** Response to current injection in MBON-γ5β'2a.

**Figure supplement 3—source data 1.** Numerical data for **Figure 3—figure supplement 3**.

---

pathways, providing additional excitatory drive to PAM DANs (**Figure 4B**). Intriguingly, SMP108 also appeared as an outstanding cell type to receive direct inputs from MBON-γ5β'2a and output to DANs (**Figure 4C**). As discussed above, we identified the γ5/β'2a as additional compartments that, like α1, can instruct second-order memory. Taken together, among other candidate cell types such as CRE011

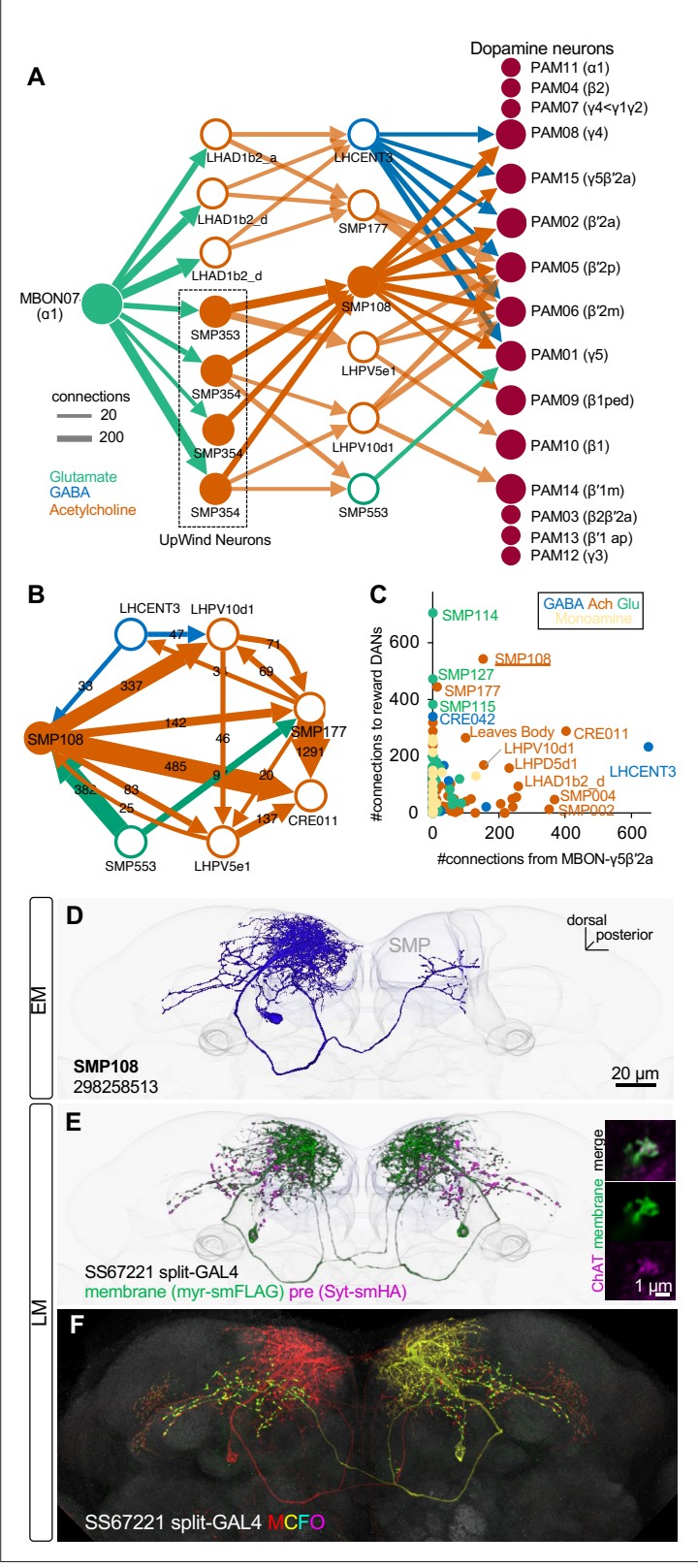

**Figure 4.** SMP108 is a key interneuron between MBON-α1 and DANs. (**A**) The connections from MBON-α1 to PAM cluster DANs with two interneurons identified in the hemibrain EM data (*Scheffer et al., 2020*). The width of arrows indicate number of connections. The colors of circles and arrows indicate type of putative neurotransmitter. Single SMP353 and three SMP354s have similar morphology and projection patterns and converge on to

*Figure 4 continued on next page*

*Figure 4 continued*

SMP108. Cholinergic interneurons SMP353/SMP354 and SMP108 are shown as filled orange circles and arrows. Other cholinergic connections are shown in transparent orange. See *Supplementary file 1* for a full connectivity matrix and neurotransmitter predictions. See *Figure 4—figure supplement 2* for the SMP108's connections with subtypes of DANs. (**B**) Connections between the six neurons in the second layer in A and CRE011. SMP108 outputs to all three other putative cholinergic interneurons. LHPV10d1 is the top target of SMP108. SMP553 send its first and second strongest outputs to SMP108 and SMP177. (**C**) Total number of connections to reward DANs (PAM01, 02, 04, 06, 07, 08, 10, 11,15) which can induce appetitive memory with optogenetic activation, plotted against number of inputs from MBON-γ5β'2a. Each circle represents one of 396 interneuron cell types that have at least 100 total connections with MBONs and DANs. Similar to SMP108, CRE011 is an outlier cell type in terms of the high number of direct inputs from MBON-γ5β'2a and outputs to reward DANs. See *Figure 4—figure supplement 3* for other kinds of connections between these interneurons and DANs/MBONs. (**D**) A projection of a reconstructed SMP108 neuron in the hemibrain EM images aligned to a standard brain with outline of the brain and the MB lobes. (**E**) Confocal microscope images of SS67221 split-GAL4 driver with membrane-targeted reporter myr-smFLAG and presynaptic reporter Syt-smHA. Inset shows anti-ChAT immunoreactivity of SMP108's axon terminals. (**F**) Morphology of individual SMP108 visualized by multi-color flip out of SS67221 split-GAL4.

The online version of this article includes the following figure supplement(s) for figure 4:

**Figure supplement 1.** Connections of MBON-α1 and SMP108.

**Figure supplement 2.** Connections from SMP108 to DAN subtypes.

**Figure supplement 3.** Connections of interneurons with DANs and MBONs.

and LHPD5d1 (*Figure 4C*), the circuit centered at SMP108 appears to be a prominent candidate that converts first-order plasticity in both α1 and γ5β'2a compartments to excitatory drive to DANs.

Identification of SMP108 and its associated circuits allowed us to construct a few testable hypotheses regarding the circuit mechanisms of second-order conditioning. First, SMP108's response to the reward-predicting S1 odor should be potentiated after first-order conditioning. Second, activation of SMP108 should trigger dopamine release in the MB compartments involved in appetitive memory. Third, the output of SMP108 should be required for second-order memory. Fourth, memory induced by the SMP108 pathway should recapitulate the transient and unstable nature of second-order memory. To experimentally test those hypotheses, we generated split-GAL4 drivers for SMP108 (SS67221 and SS45234; *Figure 4D–F*). Using these drivers, we confirmed that axonal terminals of SMP108 are immunoreactive to choline acetyltransferase (*Figure 4E*), which is consistent with the fact that 2416 out of 2753 presynaptic sites of SMP108 are predicted to be cholinergic in the hemibrain data (*Supplementary file 1*).

## SMP108 acquires enhanced response to reward-predicting odor

First, we examined the change in SMP108's odor responses after pairing of an odor and optogenetic activation of PAM-cluster DANs, which can induce appetitive memory. As expected from the converging inputs from multiple lateral horn cell types (*Supplementary file 1*), SMP108 showed robust spiking responses to odors. After pairing, responses to the paired odor were selectively potentiated (*Figure 5*). Furthermore, reversal pairing de-potentiated the previously paired odor. Thus, SMP108 is capable of acquiring enhanced responses to S1 after first-order conditioning and flexibly tracking updates of odor-reward associations.

## SMP108-evoked dopamine release in appetitive memory compartments

Next, we directly measured the pattern of dopamine release evoked by optogenetic activation of SMP108, its upstream neurons (SMP353 and SMP354), or DANs using a recently developed dopamine indicator DA2m (*Sun et al., 2020*). With direct stimulation of DANs, release of dopamine was largely restricted to the compartment(s) innervated by Chrimson-expressing DANs (*Figure 6—figure supplement 1*). Consistent with EM connectivity, activation of SMP108 or SMP353/354 evoked dopamine release in the reward memory compartments β'2, γ4 and γ5 compartments (*Figure 6*). SMP108 activation also evoked small dopamine release in β1 and β2, presumably via indirect connections, but not in α1. Notably, we observed that the dopamine signal in γ2, which is tuned to punitive stimuli, was significantly reduced after SMP108 activation (*Figure 6—figure supplement 1C*). Other DANs for aversive memories such as PAM-γ3, PPL1-γ1pedc, and PPL1-α3 showed very weak response, if

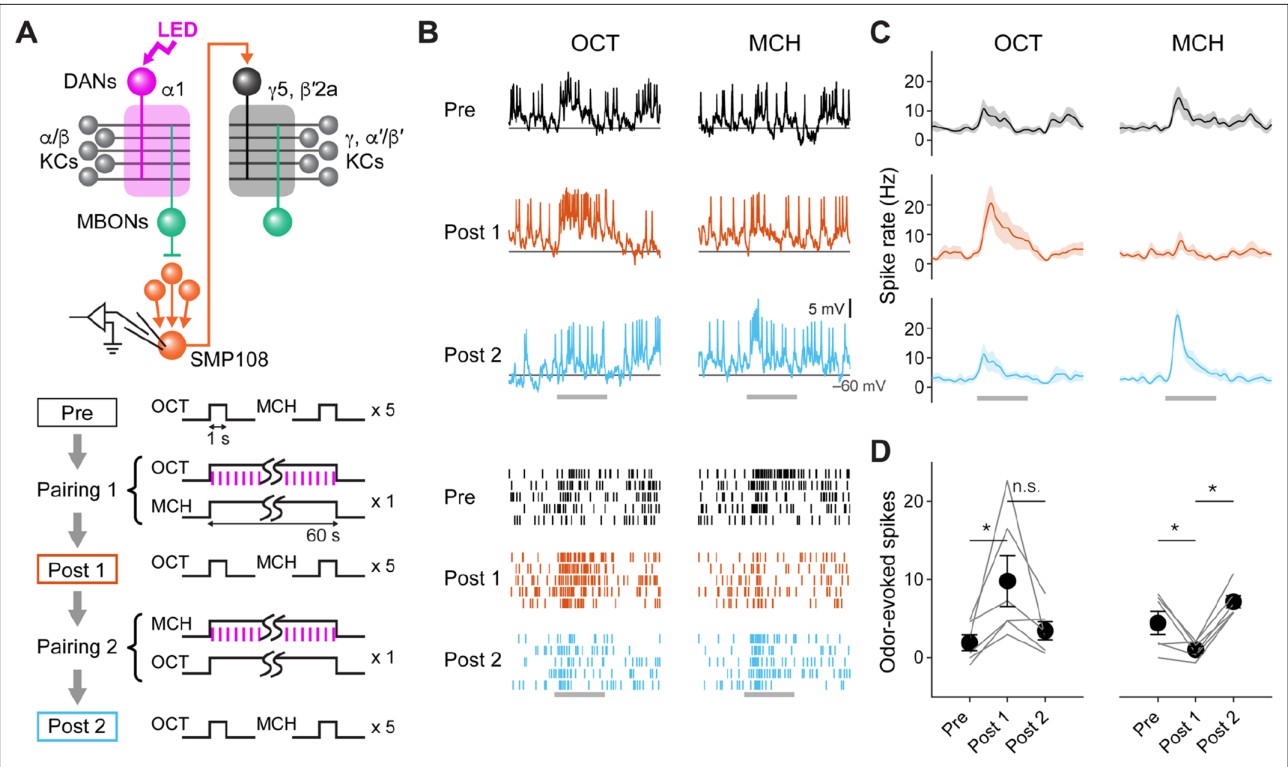

**Figure 5.** SMP108 acquires enhanced responses to reward-predicting odors. (**A**) Experimental design and protocol. ChrimsonR-mVenus was expressed in PAM-cluster DANs, which include PAM-α1, using R58E02-LexA. In vivo whole-cell recordings were made from SMP108, which was labeled by GFP using a split-GAL4 driver SS45234. In the first pairing (Pairing 1), 1 min presentation of OCT was paired with LED stimulation (1ms, 2 Hz, 120 times), followed by 1 min presentation of MCH alone. Odors were flipped in the second round of pairing (Pairing 2). Responses to each odor (1 s presentation) were measured before (Pre) and after pairing 1 (Post 1), and after pairing 2 (Post 2). (**B**) Membrane voltage (upper panels) and spike data (lower panels) from a single representative neuron. Gray bars indicate 1 s odor presentation. (**C**) Time courses of instantaneous spike rate (mean ± SEM; n=6). (**D**) Summary data of mean odor-evoked spike counts ( ± SEM). Gray lines indicate data from individual neurons. After each pairing, responses to paired odors were potentiated, while those to unpaired odors tended to decrease. Repeated-measures two-way ANOVA (p=0.0001) followed by Tukey's post hoc multiple comparisons test. *p<0.05.

The online version of this article includes the following source data for figure 5:

**Source data 1.** Numerical data for *Figure 5*.

any. Thus, activation of SMP108 triggers dopamine release selectively in multiple reward memory compartments.

## SMP108 is required for second-order conditioning

As expected from above results, we found that blocking neurotransmission of SMP108 by expression of TNT using two different split-GAL4 drivers impaired second-order conditioning compared to genetic controls (*Figure 7A*). We were unable to block SMP108 only during the second-order conditioning using the thermogenetic effector shibire[ts1] because flies with control genotype rapidly extinguished the first-order memory and failed to perform second-order conditioning at the restrictive temperature of 32°C (data not shown). Nonetheless, blocking SMP108 with TNT did not impair the first-order memory with 2 min or 1 day retention (*Figure 7B*), indicating that flies with blocked SMP108 were fully capable of smelling odors, tasting sugar, and forming, consolidating, and retrieving the first-order appetitive memory.

To further assess the potential contribution of SMP108 to appetitive memory retrieval, we tested whether activation of SMP108 triggers any relevant behavior. Flies steer to an upwind orientation in the presence of reward-predicting odors and food-related odors like vinegar (*Álvarez-Salvado et al., 2018*; *Borst and Heisenberg, 1982*; *Handler et al., 2019*). Upon optogenetic stimulation of SMP108 with CsChrimson, flies indeed changed their mean orientation and walked upwind in the same circular arena used in the olfactory conditioning experiments described above (*Figure 7—figure*

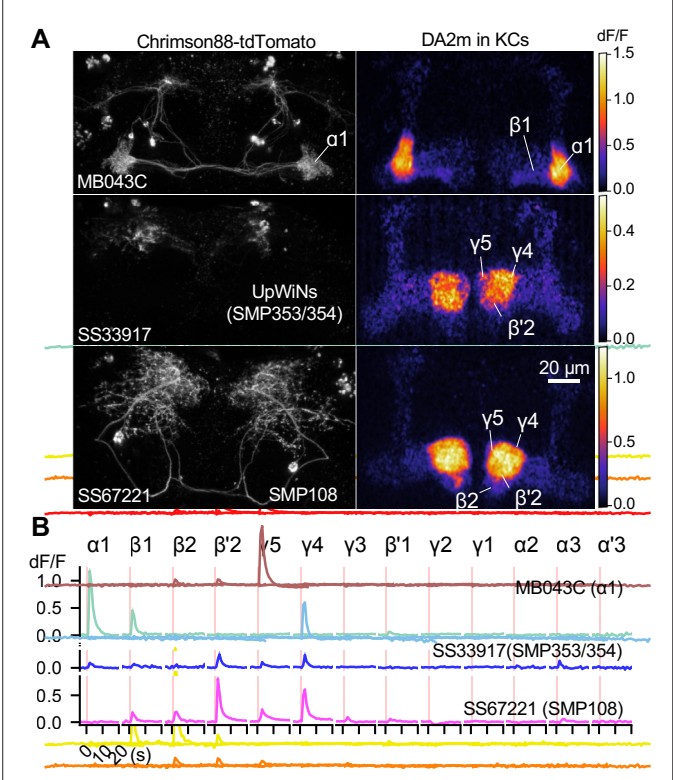

**Figure 6.** SMP108 promotes dopamine release in multiple compartments. (**A**) Representative images of Chrimson88-tdTtomato expression patterns (left) and maximum intensity projections of DA2m dF/F in the MB lobes (right). Release of dopamine upon activation of DANs or SMP108 pathways, measured with dopamine sensor DA2m expressed in Kenyon cells. *10XUAS-Syn21-Chrimson88-tdTtomato-3.1 in attP18* was driven with designated split-GAL4 driver lines. Fluorescence of DA2m in response to one second of 660 nm LED light was measured in dissected brains with two-photon imaging of volume containing MB lobes (see Materials and methods). (**B**) Mean DA2m dF/F in ROIs defined for each MB compartment. SEMs are shown as shading, although they are often within width of lines representing means. N=8–12. See *Figure 6—figure supplement 1* for quantification and the data with direct simulation of DANs.

The online version of this article includes the following source data and figure supplement(s) for figure 6:

**Figure supplement 1.** Patterns of dopamine release by different driver lines.

**Figure supplement 1—source data 1.** Numerical Data for *Figure 6*.

*supplement 1A*). However, we did not observe any impairment of upwind steering in response to the sugar-associated odor in SMP108-blocked flies (*Figure 7—figure supplement 1B*), suggesting the existence of redundant circuits that trigger memory-based upwind steering. Thus, SMP108 could contribute to retrieval of reward memory for guiding actions, but its requirement is limited to second-order conditioning. Taken together, these results indicate that SMP108, which we identified as a prominent anatomical hub for the feedforward circuit between reward memory compartments, indeed plays a key role in second-order conditioning by triggering dopamine signals in response to the reward-predicting cue.

## SMP108 pathway induces transient memory

Based on the results so far, we propose a teacher-student compartment model that explains the induction mechanism of second-order memory and its distinct dynamics from first-order memory (*Figure 8A*). In this model, local plasticity induced in a stable memory compartment (i.e. α1) during first-order conditioning functions as a reinforcer to induce secondary plasticity in other transient memory compartments through interneurons (i.e. SMP108) that connect those memory compartments. Thus, this model predicts that target compartments of SMP108 pathway collectively express

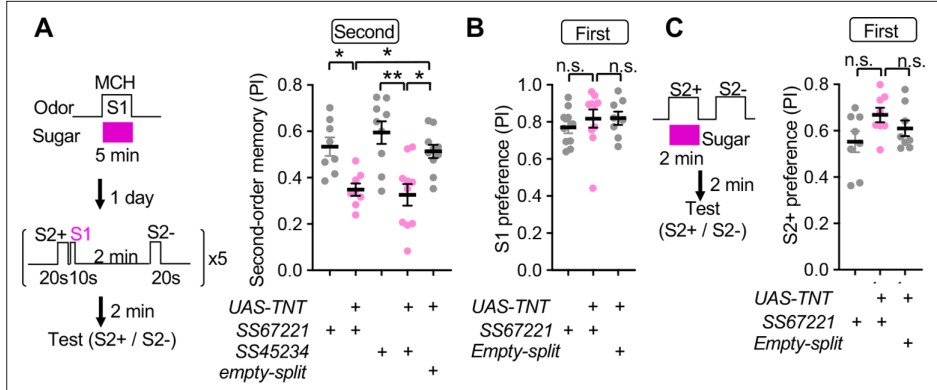

**Figure 7.** SMP108 is required for second-order memory. (**A**) Second-order memory immediately after five training sessions as in *Figure 1D* following 5-min first-order conditioning a day before. Blocking SMP108 by expressing TNT with SS67221 or SS45234 impaired the second-order memory compared to genetic controls. N=10–12. (**B**) Preference to the S1 (MCH) odor over the air one day after pairing with sugar for 5 min. N=8–10. (**C**) First-order memory immediately after pairing S2+ odor with sugar for 2 min. N=8. *, p<0.05; **, p<0.01; Dunn's multiple comparison tests following Kruskal-Wallis test.

The online version of this article includes the following source data and figure supplement(s) for figure 7:

**Source data 1.** Numerical Data for *Figure 7*.

**Figure supplement 1.** SMP108 can drive upwind steering but dispensable for the conditioned responses.

**Figure supplement 1—source data 1.** Numerical Data for *Figure 7—figure supplement 1*.

transient memory dynamics that recapitulates unstable nature of second-order memory induced by sugar-odor (*Figure 1F and G*) or optogenetic conditioning (*Figures 1H, 2E and G*).

To test this prediction, we next examined the dynamics of memory induced by the SMP108 pathway in detail and compared them to those induced by direct stimulation of PAM-α1 and other DAN types using CsChrimson (*Figure 8B* and *Figure 8—figure supplement 1*). The protocol started by assessing naïve odor preference that was designed to be canceled by reciprocal experiments. Then flies were sequentially trained five times by 10 s, 30 s, 60 s, 60 s, and 60 s periods of odor presentation paired with LED activation, and then another odor presented without LED activation (training phase). Memory was tested by giving a choice between odors after each training. After the fifth training, memory was tested 12 times without pairing with LED activation (extinction phase). Then flies were trained with a reversal protocol 5 times and tested 12 times (reversal phase). After one more round of reversal phase (re-reversal), flies were exposed to LED activation without odor to test the susceptibility of memory to non-contingent activation of DANs, a protocol that is known to erase memory (*Berry et al., 2012*; *Plaçais et al., 2012*). These experiments revealed that memories induced by SMP108 or its upstream SMP353/354 differ in several ways from the memory induced by activation of PAM-α1 (*Figure 8C–F*). First, SMP108 and SMP353/354 can induce memory more rapidly than PAM-α1 (*Figure 8C*). Second, memories formed by SMP108 and SMP353/354 declined during later training sessions and during the extinction phase, whereas memory formed by PAM-α1 remained high (*Figure 8D and E*). Third, memory formed by PAM-α1 was resistant to DAN activation, but memories formed by SMP108 and SMP353/354 were decreased (*Figure 8F*). Such transient learning and fast extinction are reminiscent of second-order conditioning by sugar (*Figure 1F and G*) or optogenetics (*Figures 1H, 2E and G*). In contrast to the activation of CsChrimson in PAM-α1, drivers that target CsChrimson to SMP108's downstream DANs exhibited memory dynamics similar to those observed when CsChrimson is activated in SMP108 or SMP353/354. For instance, MB032B and MB213B split-GAL4 that target CsChrimson in β′2m and β1/β2, respectively, induced transient memories (*Figure 8E*). Consistent with this, fitting the memory dynamics formed by SMP108 with a linear sum of direct DAN activation data indicated an overweight of MB032B (β′2m), MB213B (β1/β2) and MB312C (γ4), and zero weight for MB043C (α1) (*Figure 8G*). However, the high memory score of SMP108 activation after the first 10 s training was fitted poorly, indicating that combinatorial activation of DANs and/or suppression of DANs innervating γ2 (*Figure 6—figure supplement 1C*) might have a synergistic effect on memory formation. These experiments highlight the distinct memory

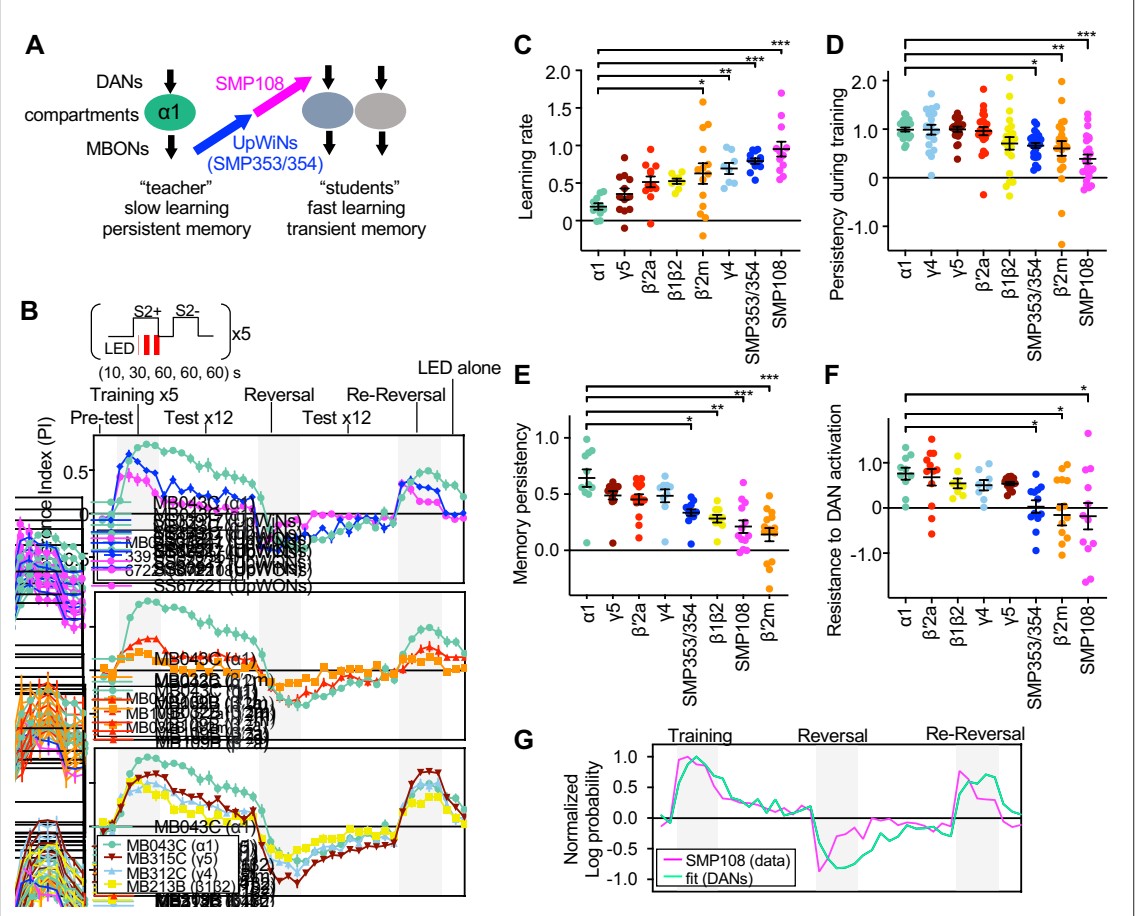

**Figure 8.** SMP108 pathway induces transient memory. (**A**) Teacher-student compartments model of second-order conditioning hypothesizes that 'teacher' compartment with slow learning rate and persistent memory instructs other compartments with faster learning rate and transient memory dynamics via SMP353/SMP354 and SMP108. (**B**) Dynamics of memory with optogenetic activation of SMP108 (SS67221), SMP353/354 (SS33917) or various types of DANs. See texts and Materials and methods for explanation of the protocol, and *Figure 8—figure supplement 1* for specificity of expression pattern in the central brain and the ventral nerve cord. Means and SEM are displayed. N=8–14. (**C**) Learning rate defined as a (PI after first 10 s training)/(peak PI during the first 5 training trials) for each driver line. (**D**) Persistency during training defined as (PI after 5th training)/(peak PI during the first 5 training trials). (**E**) Persistency of memory defined as (mean of PIs during 12 tests after first training trials)/(peak PI during the first 5 x training trials). (**F**) Resistance to DAN activation defined as (mean of last three tests following activation LED without odors)/(PI after 5th conditioning in re-reversal phase), which measures both transiency during training and extinction during 12 tests. p<0.05; **, p<0.01; ***, p<0.01; Dunn's multiple comparison tests following Kruskal-Wallis test; N=8–14. (**G**) The log-probability ratio of choosing the S2+ against S2− for SS67221 (SMP108) data were fitted best with weights of (0.57, 0.46,0.157,0,0,0) for data of DAN driver lines (MB032B, MB213B, MB312C, MB043C, MB109B, and MB315C).

The online version of this article includes the following source data and figure supplement(s) for figure 8:

**Source data 1.** Numerical Data for *Figure 8*.

**Figure supplement 1.** Expression patterns of drivers.

properties exhibited by upstream and downstream partners of SMP108, and might help explain the circuit mechanisms underlying the difference between first- and second-order memories.

## Discussion

In this study, we used the *Drosophila* mushroom body as a model system to examine how multiple dopamine-driven memory circuits interact to enable second-order conditioning. Although second-order conditioning has been demonstrated behaviorally in many species, there is little circuit-level knowledge to provide mechanistic insight. By developing a robust appetitive second-order conditioning protocol and utilizing the EM connectome map in *Drosophila*, we uncovered neural circuit mechanisms that define dynamics and learning rules of second-order conditioning.

## Origins of the unique learning rules of second-order conditioning

Our optimization of the second-order conditioning protocol using actual sugar reward or its optogenetic substitution revealed important properties of second-order memory and enabled detailed circuit interrogation. Formation of second-order memory was most effective either when the first-order S1 odor predicted a strong sugar reward (*Figure 1D*) or when long-term first-order memory was optogenetically induced (*Figure 2B*). Furthermore, during second-order training following optogenetic first-order conditioning, S2 odor must precede the S1 odor (*Figure 2C*). With additional second-order training sessions, second-order memory could become as robust as the first-order memory, but the continual omission of the expected fictive reward during training and extinction trials tended to reduce second-order memory (*Figures 1H, 2E and G*). The retention of second-order memory was also shorter than first-order memory when we used actual sugar reward for first-order conditioning (*Figure 1F*). Remarkably, all the dynamics and learning rules we found in *Drosophila* for second-order conditioning are well-conserved across animal phyla (*Gewirtz and Davis, 2000*; *Pavlov and Gantt, 1927*; *Rescorla, 1980*). Our study indicates that, in flies, at least some of these phenomena can be accounted for by the teacher-student model of the MB circuit, which hypothesizes distinct dynamics of plasticity in individual compartments and hierarchical interactions between compartments. Namely, a compartment with a slow learning rate instructs compartment(s) with transient memory dynamics.

Requirement of long first-order training for successful formation of second-order memory (*Figure 1C and D*) can be explained by the properties of the α1, which we identified as the teacher compartment. The DANs in α1 respond to sugar relatively weakly compared to other DANs in the β′2, β2, γ4, γ5 compartments (*Siju et al., 2020*). Also the α1 compartment exhibited the slowest learning rate of all compartments even with optogenetic stimulation of DANs that efficiently release dopamine (*Figure 6—figure supplement 1* and *Figure 8C*). Once established, however, memory in the α1 is highly resistant to extinction (*Figures 2A and 8D*), which is likely critical for forming second-order conditioning without compromising first-order memory. These considerations emphasize the eligibility of the α1 compartment as a teaching compartment among all reward-memory compartments. On the other hand, transient and unstable nature of second-order memory can be ascribed to collective properties of student compartments (*Figure 8*). Future studies are required to identify intrinsic molecular factors and microcircuit elements responsible for distinct dynamics of teacher and student compartments.

## Implications to the higher-order functions of heterogeneous dopamine subsystems

Our study identified a role of hierarchical interaction between dopamine-based memory subsystems. Importantly, heterogeneous populations of DANs are also found in vertebrate species, and they are involved in distinct types of learning. Studies using visual conditioning in monkeys found that distinct types of DANs projecting to the head or tail regions of the caudate nucleus change their response to reward-predicting cues with very different dynamics (*Kim et al., 2015*; *Kim et al., 2014*). A recent study in rodents indicated that subsets of DANs have diverse learning rates to compute positive and negative reward prediction errors to enable distributional reinforcement learning (*Dabney et al., 2020*). Cue-evoked dopamine transients at the onset of reward-predicting cues are required for second-order conditioning in rodents (*Maes et al., 2020*). Such dopamine transients could be derived from memory encoded by the same DAN, other type(s) of DANs, or both, depending on the architecture of feedback circuits. Given the conserved nature of second-order memory transiency across animal phyla, future studies in vertebrate models may also reveal a hierarchical interaction between dopamine cell types with fast and slow dynamics in second-order conditioning.

Second-order conditioning is merely one example of learning that depends on higher-order connections between dopamine-dependent memory subsystems. In fact, in flies, feedback and feedforward connections between MBONs and DANs or lateral connections between MBONs are implicated in extinction of aversive and appetitive memory as well as consolidation of memories (*Felsenberg et al., 2018*; *Felsenberg et al., 2017*; *McCurdy et al., 2021*). The EM connectome map, along with computational modeling (*Gkanias et al., 2022*; *Jiang et al., 2021*), will guide further investigation of intercompartmental interactions. For instance, we identified one outlier cell type of GABAergic interneuron LHCENT3 that receives inputs from glutamatergic MBON-γ5β′2a and outputs to reward DANs (*Figure 4C*). This cell type may serve as the substrate for subtraction of expected reward in

the computation of reward prediction error, as GABAergic neurons in VTA do in vertebrate brains (*Starkweather and Uchida, 2021*). Although the majority of circuit-level research has focused on rather simple forms of learning that involve primary reinforcers, animals have abundant opportunities to shape their behaviors through indirect learning that depends on existing memory. We expect that network motifs similar to what we identified here contribute to various forms of such complex learning. We expect that future modeling studies constrained by the EM connectome and large-scale behavioral and neural activity data will lead to a comprehensive understanding of the MB's contributions to these computations.

## Contents of second-order conditioning

Understanding what is learned is a fundamental challenge in studies of associative learning. There are many possible structures of associations that would allow animals to perform second-order conditioning tasks. Our finding of the cross-compartmental nature of second-order conditioning makes it unlikely that flies associate S2 with a *specific* type of reward used as US, because individual MB compartments are tuned to different kinds of rewards or reward responses. That is, while DANs in the teacher compartment α1 are essential for nutritional value learning (*Yamagata et al., 2015*), those in the student compartments γ4 and β′2 respond to water in thirsty flies (*Lin et al., 2014*). DANs in γ4, γ5 and β′2 also represent vinegar and activity of DANs in γ4 correlates with upwind steering (*Lewis et al., 2015*; *Zolin et al., 2021*). DANs in β′2a also respond to a punishment-predicting odor when punishment is omitted (*McCurdy et al., 2021*). Thus, based on our circuit mapping and the known functions of the relevant circuits, we propose that S2 is associated with positive valence that was originally associated with S1 but generalized to broader types of rewards. This view is consistent with the fact that second-order conditioning is typically insensitive to subsequent reduction of the value of the US (i.e. devaluation), which suggests that an association is formed between S2 and the original valence of the US rather than the US itself (*Rescorla, 1980*). Studies in rodents demonstrated that S1 and S2 with different sensory modalities can elicit distinct conditioned responses (CRs), supporting the idea that S2 is not associated with the specific CR elicited by S1 (*Holland, 1977*; *Kim et al., 1996*). Notably, a broadening of the category of expected rewards in second-order conditioning has been suggested by a study in pigeons (*Stanhope, 1992*), where differential CRs to qualitatively distinct USs (i.e. food and water) were observed for S1 but not for S2. Thus, our circuit underpinning of second-order conditioning provides a concrete neuronal substrate for behavioral and psychological phenomena that have been described for decades.

## Materials and methods

### Fly strains

*Drosophila melanogaster* strains were reared at 22 °C and 60% humidity on standard cornmeal food in 12:12 hr light:dark cycle. Four to 10 days of adult females were used 2–4 days after sorting them on the Peltier cold plate. For flies expressing Chrimson (*Klapoetke et al., 2014*) the food was supplemented with retinal (0.2 mM all-trans-retinal prior to eclosion and then 0.4 mM). Driver and effector lines are listed in the key resource table and genotypes used by each figure are listed below. The new collection of split-GAL4 and split-LexA drivers was designed based on confocal image databases (http://flweb.janelia.org) (*Jenett et al., 2012*), and screening expression patterns of p65ADZp and ZpGAL4DBD combinations as described previously (*Aso et al., 2014*; *Pfeiffer et al., 2010*). Confocal stacks of new split-GAL4 driver lines used in this study are available at http://www.janelia.org/split-gal4.

### Detailed fly genotypes used by figures

| Figure | Genotype |
| --- | --- |
| *Figure 1C–G*, *Figure 1—figure supplement 1* | *Canton S* |

*Continued on next page*

*Continued*

| Figure | Genotype |
|---|---|
| *Figure 1H* | *w/w, 20xUAS-CsChrimson-mVenus attP18;+/Gr64f-GAL4;+/Gr64f-GAL4* |
| *Figure 2A–C* | *w/w, 20xUAS-CsChrimson-mVenus attP18;;+/MB043C-split-GAL4*<br>*w/w, 20xUAS-CsChrimson-mVenus attP18;+/MB213B-split-GAL4*<br>*w/w, 20xUAS-CsChrimson-mVenus attP18;;+/MB312C-split-GAL4*<br>*w/w, 20xUAS-CsChrimson-mVenus attP18;MB109B/MB315C-split-GAL4*<br>*w/w, 20xUAS-CsChrimson-mVenus attP18;+/ Empty-split-GAL4* |
| *Figure 2D* | *w/+;Empty-split-GAL4/UAS-TNT (II)*<br>*w/+;MB196B/UAS-TNT (II)*<br>*w/+;MB043C/UAS-TNT (II)* |
| *Figure 3*<br>*Figure 3—figure supplement 3* | *w/w,13XLexAop2-IVS-ChrimsonR-mVenus-p10 attP18, 20XUAS-syn21 mScarlet-opt-p10 su(Hw)attp8; SS01308-split-GAL4/MB043-split-LexA* |
| *Figure 3—figure supplement 1* | *w/w,13XLexAop2-IVS-ChrimsonR-mVenus-p10 attP18, 20XUAS-syn21 mScarlet-opt-p10 su(Hw)attp8; +/MB043-split-LexA*<br>*w/w,13XLexAop2-IVS-ChrimsonR-mVenus-p10 attP18, 20XUAS-syn21 mScarlet-opt-p10 su(Hw)attp8; MB319C-split-GAL4/MB043-split-LexA*<br>*w/w,13XLexAop2-IVS-ChrimsonR-mVenus-p10 attP18, 20XUAS-syn21 mScarlet-opt-p10 su(Hw)attp8; SS01308-split-GAL4/MB043-split-LexA*<br>*w/w,13XLexAop2-IVS-ChrimsonR-mVenus-p10 attP18, 20XUAS-syn21 mScarlet-opt-p10 su(Hw)attp8; SS67221-split-GAL4/MB043-split-LexA* |
| *Figure 3—figure supplement 2* | *w/w,13XLexAop2-IVS-ChrimsonR-mVenus-p10 attP18, 20XUAS-syn21 mScarlet-opt-p10 su(Hw)attp8; MB319C-split-GAL4/MB043-split-LexA* |
| *Figure 4E* | *w/w, pJFRC200-10xUAS-IVS-myr::smGFP-HA in attP18; pJFRC225-5xUAS-IVS-myr::smGFP-FLAG in VK00005/SS67221-split-GAL4* |
| *Figure 4F* | *pBPhsFlp2::PEST in attP3;; pJFRC201-10XUAS-FRT>STOP > FRT-myr::smGFP-HA in VK0005, pJFRC240-10XUAS- FRT >STOP > FRT-myr::smGFP-V5-THS-10XUAS-FRT>STOP > FRT-myr::smGFP-FLAG in su(Hw)attP1/SS67221-split-GAL4* |
| *Figure 5* | *13XLexAop2 IVS p10 ChrimsonR mVenus trafficked in attP18/+; 58E02-LexAp65 in attP40 / VT026646-p65ADZp in attP40 (ss45234-split); pJFRC28-10XUAS-IVS-GFP-p10 in su(Hw) attP1 /VT029309-ZpGdbd in attP2 (ss45234-split)* |
| *Figure 6*,<br>*Figure 6—figure supplement 1* | *w/w, 10XUAS-Chrimson88-tdTomato attP18; 13F02-LexAp65 attP40; LexAop2-DA2m VK00005/MB043C-split-GAL4*<br>*w/w, 10XUAS-Chrimson88-tdTomato attP18; 13F02-LexAp65 attP40; LexAop2-DA2m VK00005/MB213B-split-GAL4*<br>*w/w, 10XUAS-Chrimson88-tdTomato attP18; 13F02-LexAp65 attP40; LexAop2-DA2m VK00005/MB032B-split-GAL4*<br>*w/w, 10XUAS-Chrimson88-tdTomato attP18; 13F02-LexAp65 attP40; LexAop2-DA2m VK00005/MB109B-split-GAL4*<br>*w/w, 10XUAS-Chrimson88-tdTomato attP18; 13F02-LexAp65 attP40; LexAop2-DA2m VK00005/MB315C-split-GAL4*<br>*w/w, 10XUAS-Chrimson88-tdTomato attP18; 13F02-LexAp65 attP40; LexAop2-DA2m VK00005/MB312C-split-GAL4*<br>*w/w, 10XUAS-Chrimson88-tdTomato attP18; 13F02-LexAp65 attP40; LexAop2-DA2m VK00005/SS33917-split-GAL4*<br>*w/w, 10XUAS-Chrimson88-tdTomato attP18; 13F02-LexAp65 attP40; LexAop2-DA2m VK00005/SS67221-split-GAL4* |
| *Figure 7* | *w/+;SS67221/+*<br>*w/+; SS67221/UAS-TNT (II)*<br>*w/+;SS45234/+w/+; SS45234/UAS-TNT (II)*<br>*w/+;Empty-split-GAL4/TNT (II)SS67221/TNT* |
| *Figure 7—figure supplement 1A* | *w/w, 20xUAS-CsChrimson-mVenus attP18;+/ Empty-split-GAL4*<br>*w/w, 20xUAS-CsChrimson-mVenus attP18;+/SS67221-split-GAL4* |
| *Figure 7—figure supplement 1B* | *w/+;SS67221/+*<br>*w/+; SS67221/UAS-TNT (II)*<br>*w/+;Empty-split-GAL4/TNT (II)SS67221/TNT* |

*Continued on next page*

*Continued*

| Figure | Genotype |
|---|---|
| *Figure 8*, *Figure 8—figure supplement 1* | *w/w, 20xUAS-CsChrimson-mVenus attP18;+/+;+/MB043C-split-GAL4* <br> *w/w, 20xUAS-CsChrimson-mVenus attP18;+/SS33917-split-GAL4* <br> *w/w, 20xUAS-CsChrimson-mVenus attP18;+/SS67221-split-GAL4* <br> *w/w, 20xUAS-CsChrimson-mVenus attP18;+/MB032B-split-GAL4* <br> *w/w, 20xUAS-CsChrimson-mVenus attP18;+/MB109B-split-GAL4* <br> *w/w, 20xUAS-CsChrimson-mVenus attP18;+/+;+/MB315C-split-GAL4* <br> *w/w, 20xUAS-CsChrimson-mVenus attP18;+/MB312C-split-GAL4* <br> *w/w, 20xUAS-CsChrimson-mVenus attP18;+/MB213B-split-GAL4* |

## Olfactory conditioning

Olfactory conditioning was performed as previously described (*Aso and Rubin, 2016*). Groups of approximately 20 females of 4–10 days post-eclosion were trained and tested using the modified four-field olfactory arena (*Aso and Rubin, 2016*; *Pettersson, 1970*) equipped with the 627 nm LED board (34.9 µW/mm2 at the position of the flies) and odor mixers. The flow rate of input air from each of the four arms was maintained at 100 mL/min throughout the experiments by mass-flow controllers, and air was pulled from the central hole at 400 mL/min. Odors were delivered to the arena by switching the direction of airflow to the tubes containing diluted odors using solenoid valves. The odors were diluted in paraffin oil: 3-octanol (OCT 1:1000), 4-methylcyclohexanol (MCH; 1:750), Pentyl acetate (PA: 1:10000) and ethyl lactate (EL: 1:10000). Sugar conditioning was performed by using tubes with sucrose absorbed Whatman 3 MM paper as previously described (*Krashes and Waddell, 2008*; *Liu et al., 2012*). Before conditioning, flies were starved for 40–48 hr on 1% agar. Videography was performed at 30 frames per second and analyzed using Fiji. For experiments with one day retention, flies were kept in agar vials at 21 °C after first-order conditioning. For testing olfactory memories, distribution of flies in four quadrants were measured for 60 s. The performance index (PI) is defined as a mean of [(number of flies in the two diagonal quadrants filled the one odor) - (number of flies in other two quadrants filled with another odor or air)]/(total number of flies) during final 30 s of 60 s test period. The average PI of reciprocal experiments is shown in figures to cancel out potential position bias and innate odor preference. Although genotypes of flies were not hidden to experimentalists, handling was minimized by automation of stimulus delivery. We included all the data if experiments were validated by metadata such as airflow readout from the mass flow controllers.

## Optimization of second-order conditioning

To establish a training protocol for robust olfactory second-order conditioning in *Drosophila*, we first characterized how innate preference for an odor (when compared with pure air) changes over multiple trials using the four-armed olfactory arena (Figure-figure supplement 1) (*Aso and Rubin, 2016*; *Pettersson, 1970*). We previously chose concentrations of two conventional odors, 4-methylcyclohexanol (MCH) and 3-octanol (OCT), so that naïve fed flies show behavioral responses to each odor at a similar level, minimizing bias between them (*Tully and Quinn, 1985*). At the same concentration, starved flies showed slight attraction to the MCH at the first trial, then gradually shifted to aversion in subsequent trials (*Figure 1—figure supplement 1*). In contrast, both fed and starved flies showed aversion to the OCT, which gradually decreased in subsequent trials. Because the innate aversiveness of OCT may preclude appetitive second-order conditioning, we decided to use MCH as the first conditioned stimulus (S1) throughout this study.

The strength of second-order conditioning tends to be low, compared to that of first-order, but can be enhanced by using an unconditioned stimulus (US) of high intensity and sensory stimuli within the same modality (*Helmstetter and Fanselow, 1989*; *Rescorla and Furrow, 1977*). Thus, we examined the effect of increasing conditioning duration. After pairing MCH with sugar for increasing durations (0, 2, 5 min), flies were allowed to consolidate the memory for one day. Then the stability of first-order memory was tested by repeating binary choice between S1 odor and air for 12 times. All trained flies showed attraction to MCH during at least the first five trials (*Figure 1C*). One 2 min training was enough to induce appetitive memory (*Krashes and Waddell, 2008*; *Tempel et al., 1983*), but longer 5 min training resulted in slightly stronger memories during the first five tests on average. Therefore, we decided to limit the number of second-order conditioning to five times. We used two odorants, pentyl acetate (PA) and ethyl lactate (EL) as the second conditioned stimuli (S2). These odors are

known to evoke discrete patterns of activity in Kenyon cells (*Campbell et al., 2013*) and thought to be easily discriminated against. Innate behavioral responses to these odors were relatively stable over 12 trials (*Figure 1—figure supplement 1*).

For first-order conditioning, flies learn best when sensory cues precede US or DAN activation (*Aso and Rubin, 2016*; *Tanimoto et al., 2004*). Thus, during second-order conditioning, 20 s of one S2 odor (S2+) was immediately followed by 10 s of the S1 odor, whereas another S2 odor (S2−) was presented alone. Flies failed to form second-order memory when S1 preceded S2+ (*Figure 2C*). PA and EL were S2+ and S2− odors, respectively, in half of a set of reciprocal experiments. The S2+ and S2− odors were swapped in the other half of reciprocal experiments. After five training sessions, unpaired control flies showed weak attraction to S2+, possibly due to innate attractiveness of MCH in starved flies (*Figure 1—figure supplement 1*). Compared to this basal response, flies preferred the S2+ odor over the S2− odor when first-order conditioning was long enough (i.e. 5 min; *Figure 1D*). This preference for the S2+ odor was not due to stimulus generalization of S1 (MCH) to PA or EL, because such bias is designed to be canceled by our experimental design involving reciprocal experiments. Both immediate and 1-day first-order memories were potent to induce second-order memory, but second-order memory did not last for one day (*Figure 1F*).

## Response airflow

For testing airflow directional response, we used the same circular olfactory arena (*Figure 7—figure supplement 1*), in which air flows from peripheral to a hole at the center. Each fly's distance from center ($r_i$) was measured and area normalized index ($r_i/r_{arena}$)*($r_i/r_{arena}$) was calculated. $r_{arena}$ is the radius of the arena. When flies distribute randomly in the arena, mean r is $1/\sqrt{2}$ and area normalized index is 1/2. To calculate upwind displacement, the mean of arena normalized distance from center at each time point in each movie was subtracted by that at the onset of LED or odor.

## Electrophysiology

Fly stocks for electrophysiological experiments were maintained at room temperature on conventional cornmeal-based medium (Archon Scientific). Experimental flies were collected on the day of eclosion, transferred to all-trans-retinal food (0.5 mM) and kept in the dark for 48–72 hr. For second-order conditioning experiments, flies were starved for 60–72 hr after feeding retinal food.

In vivo whole-cell recordings were performed as previously reported (*Hige et al., 2015*). The patch pipettes were pulled for a resistance of 4–6 MΩ and filled with pipette solution containing (in mM): L-potassium aspartate, 140; HEPES, 10; EGTA, 1.1; $CaCl_2$, 0.1; Mg-ATP, 4; Na-GTP, 0.5 with pH adjusted to 7.3 with KOH (265 mOsm). The preparation was continuously perfused with saline containing (in mM): NaCl, 103; KCl, 3; $CaCl_2$, 1.5; $MgCl_2$, 4; $NaHCO_3$, 26; N-tris(hydroxymethyl) methyl-2-aminoethane-sulfonic acid, 5; $NaH_2PO_4$, 1; trehalose, 10; glucose, 10 (pH 7.3 when bubbled with 95% $O_2$ and 5% $CO_2$, 275 mOsm). For recordings from starved flies, trehalose and glucose were replaced by equimolar sucrose. Whole-cell recordings were made using the Axon MultiClamp 700B amplifier (Molecular Devices). Target cells were visually targeted by fluorescence signal with a 60 X water-immersion objective (LUMPlanFl/IR; Olympus) attached to an upright microscope (OpenStand; Prior Scientific). Cells were held at around –60 mV by injecting hyperpolarizing current, which was typically <100 pA. Signals were low-pass filtered at 5 kHz and digitized at 10 kHz.

For odor delivery, a previously described custom-designed device was used (*Hige et al., 2015*). Saturated head space vapors of pure chemicals were air-diluted to 0.5% (for second-order conditioning) or 2% (for the other experiments) before being presented to flies. Photostimulation was delivered by a high-power LED source (LED4D067; Thorlabs) equipped with 625 nm LED. Light pulses controlled by an LED driver (DC4100; Thorlabs) were presented to the brain at 17 mW/mm² through the objective lens.

Data acquisition and analyses were done by custom scripts in MATLAB (MathWorks). Instantaneous spike rates were calculated by convolving spikes with a Gaussian kernel (SD = 50ms). Subthreshold odor responses and odor-evoked spikes were calculated with the time window of 1.2 s (for 1 s odor presentation) or 20.6 s (for 20 s odor presentation) from odor onset. Spontaneous spikes were subtracted to calculate odor-evoked spikes.

## Dopamine imaging

Virgin females of *10XUAS-Chrimson88-tdTomato attP18; R13F02-LexAp65 in attP40;LexAop2-DA2m in VK00005* (*Klapoetke et al., 2014*; *Sun et al., 2020*) were crossed with split-GAL4 driver lines, and progenies were reared at 25 °C on retinal supplemented (0.2 mM) cornmeal medium that was shielded from light. All experiments were performed on female flies, 3–7 days after eclosion. Brains were dissected in a saline bath (103 mM NaCl, 3 mM KCl, 2 mM $CaCl_2$, 4 mM $MgCl_2$, 26 mM $NaHCO_3$, 1 mM $NaH_2PO_4$, 8 mM trehalose, 10 mM glucose, 5 mM TES, bubbled with 95% $O_2$/5% $CO_2$). After dissection, the brain was positioned anterior side up on a coverslip in a Sylgard dish submerged in 3 ml saline at 20 °C. The sample was imaged with a resonant scanning 2-photon microscope with near-infrared excitation (920 nm, Spectra-Physics, INSIGHT DS DUAL) and a 25×objective (Nikon MRD77225 25XW). The microscope was controlled using ScanImage 2016 (Vidrio Technologies). Images were acquired over a 231 µm × 231 µm x 42 µm volume with a step size at 2 µm. The field of view included 512×512 pixel resolution taken at approximately 1.07 Hz frame rate. The excitation power during imaging was 19 mW.

For the photostimulation, the light-gated ion channel CsChrimson was activated with a 660 nm LED (M660L3 Thorlabs) coupled to a digital micromirror device (Texas Instruments DLPC300 Light Crafter) and combined with the imaging path with a FF757-DiO1 dichroic (Semrock). On the emission side, the primary dichroic was Di02-R635 (Semrock), the detection arm dichroic was 565DCXR (Chroma), and the emission filters were FF03-525/50 and FF01-625/90 (Semrock). An imaging session started with a 30 s baseline period, followed by a 1 s stimulation period when 12 µW/mm² photostimulation light was delivered, and responses were detected over a 30 s post stimulation period. This was repeated for 10 trials. The light intensity was measured using the Thorlabs S170C power sensor.

For quantification of dopamine sensor signals, we used custom python scripts to draw ROIs corresponding to mushroom body compartments on maximum intensity projection over time. Before calculating the change in fluorescence (ΔF), fluorescence from a background ROI was subtracted. The background ROI was drawn in a region with no fluorescence. Baseline fluorescence is the mean fluorescence over a 30 s time period before stimulation started. The ΔF was then divided by baseline to normalize signal (ΔF/F). The mean responses from the 10 trials were calculated for each animal (4–6 samples per driver). Kruskal-Wallis H (KW) test was used for multi-comparison. Post-hoc pairwise comparison was made with the Wilcoxon rank-sum test.

## Connectivity analysis

For producing the connectivity data shown in *Figure 4* and *Figure 4—figure supplements 1–3*, connectivity information was retrieved from neuPrint (https://neuprint.janelia.org/) hosting the 'hemibrain' dataset (*Scheffer et al., 2020*), which is a publicly accessible web site (https://doi.org/10.25378/janelia.12818645.v1). For cell types, we cited cell type assignments reported in *Scheffer et al., 2020*. Only connections of the cells in the right hemisphere were used due to incomplete connectivity in the left hemisphere (*Zheng et al., 2018*). Connectivity data was then imported to a software Cytoscape (https://cytoscape.org/) for generating the diagrams before finalizing on Illustrator. The 3D renderings of neurons presented were generated using the visualization tools of NeuTu (*Zhao et al., 2018*) or VVD viewer (https://github.com/takashi310/VVD_Viewer; *Kawase, 2023*; *Wan et al., 2012*).

## Neurotransmitter prediction

The method for neurotransmitter prediction using electron microscopy images and a 3D VGG-style network were described in detail for the FAFB data of a whole fly brain (*Eckstein et al., 2020*; *Zheng et al., 2018*). We used the same approach to train the network to classify individual presynaptic sites of FIB-SEM hemibrain data into the same six major neurotransmitters in fly brains as for FAB, that is: GABA, glutamate, acetylcholine, serotonin, dopamine and octopamine. Due to the differences in resolution between FAFB and the electron microscopy images used here, we adapted the architecture of the 3D VGG network to be isotropic as follows: We use four downsampling layers with uniform pooling sizes of 2x2 × 2 on 3D crops centered on synapses with a side-length of 80 voxels. The results for 396 major interneurons are summarized in *Supplementary file 1*.

## Immunohistochemistry

Brains and ventral nerve cord of 4–10 days old female were dissected, fixed and immunolabeled as previously described using the antibodies listed in the Key Resource Table (*Aso et al., 2014*; *Nern et al., 2015*). Samples were imaged with confocal microscopes (Zeiss LSM710, LSM780 or LSM880). Inset images in *Figure 4E* were taken with Airyscan.

## Regression analysis of SMP108 memory dynamics

For each strain, the log-probability ratio of reinforced vs. unreinforced stimuli was computed as $R = log\left(p/\left(1-p\right)\right)$, where $p$ is the probability of choosing the reinforced stimulus. To relate the memory dynamics induced by SMP108 to those induced by DANs that it activates, we performed non-negative linear least-squares regression of the log-probability ratio for SMP108 against the ratios for PAM DANs. This reflects an assumption that the combinatorial activation of multiple compartments contributes a behavioral bias that is additive in log-probability ratio.

## Statistics

Statistical comparisons were performed on GraphPad Prism or MATLAB using the Kruskal Wallis test followed by Dunn's post-test for multiple comparison, t-tests, or two-way ANOVA followed by Tukey's post hoc multiple comparisons test designated in figure legends. Non-parametric test was preselected for behavioral assays due to expected lack of normality or equal variance in subsets of data. Sample size was not predetermined based pilot experiments.

## Acknowledgements

We thank Daisuke Hattori, James Fitzgerald, Sandro Romani, Gerald M Rubin, Yichun Shuai, Mehrab Modi, Zongwei Chen, Adithya Rjagopalan and members of the YA and TH labs for valuable discussion and comments on the manuscript. We thank Jinyan Liu and all the members of Janelia Flylight, jET, fly facility and scientific computing for construction of behavioral setup, generation and confocal microscopy images of split-GAL4 drivers. DY was supported by Toyobo Biotechnology Foundation Postdoctoral Fellowship and Japan Society for the Promotion of Science Overseas Research Fellowship. YA was supported by HHMI. TH was supported by NIH (R01DC018874), NSF (2034783), BSF (2019026) and UNC Junior Faculty Development Award. AL-K was supported by the Burroughs Wellcome Foundation, the Gatsby Charitable Foundation, the McKnight Endowment Fund, the Simons Collaboration on the Global Brain, NIH award R01EB029858, and NSF award DBI-1707398.

# Additional information

### Funding

| Funder | Grant reference number | Author |
| --- | --- | --- |
| National Institutes of Health | R01DC018874 | Toshihide Hige |
| National Science Foundation | 2034783 | Toshihide Hige |
| BSF | 2019026 | Toshihide Hige |
| UNC Junior Faculty Development Award | | Toshihide Hige |
| Burroughs Wellcome Fund | | Ashok Litwin-Kumar |
| Gatsby Charitable Foundation | | Ashok Litwin-Kumar |
| McKnight Endowment Fund | | Ashok Litwin-Kumar |
| Simons Foundation | Simons Collaboration on the Global Brain | Ashok Litwin-Kumar Yoshinori Aso |

| Funder | Grant reference number | Author |
|---|---|---|
| National Institutes of Health | R01EB029858 | Ashok Litwin-Kumar |
| National Science Foundation | DBI-1707398 | Ashok Litwin-Kumar |
| Toyobo Biotechnology Foundation | Postdoctoral Fellowship | Daichi Yamada |
| Japan Society for the Promotion of Science | Overseas Research Fellowship | Daichi Yamada |
| Howard Hughes Medical Institute | | Daniel Bushey<br>Feng Li<br>Karen L Hibbard<br>Megan Sammons<br>Jan Funke<br>Yoshinori Aso |

The funders had no role in study design, data collection and interpretation, or the decision to submit the work for publication.

## Author contributions

Daichi Yamada, Formal analysis, Investigation, Visualization, Methodology, Writing – original draft, Writing – review and editing; Daniel Bushey, Software, Formal analysis, Investigation, Visualization, Writing – original draft, Writing – review and editing; Feng Li, Data curation, Software, Formal analysis, Visualization; Karen L Hibbard, Resources; Megan Sammons, Data curation, Visualization; Jan Funke, Data curation, Software, Formal analysis, Funding acquisition, Investigation, Methodology; Ashok Litwin-Kumar, Software, Formal analysis, Supervision, Funding acquisition, Investigation, Visualization, Writing – original draft, Writing – review and editing; Toshihide Hige, Conceptualization, Software, Formal analysis, Supervision, Funding acquisition, Validation, Investigation, Visualization, Methodology, Writing – original draft, Project administration, Writing – review and editing; Yoshinori Aso, Conceptualization, Resources, Data curation, Software, Formal analysis, Supervision, Validation, Investigation, Visualization, Methodology, Writing – original draft, Project administration, Writing – review and editing

## Author ORCIDs

Daichi Yamada ⓘ http://orcid.org/0000-0003-4944-1950
Daniel Bushey ⓘ http://orcid.org/0000-0001-9258-6579
Feng Li ⓘ http://orcid.org/0000-0002-6658-9175
Karen L Hibbard ⓘ http://orcid.org/0000-0002-2001-6099
Megan Sammons ⓘ http://orcid.org/0000-0003-4516-5928
Jan Funke ⓘ http://orcid.org/0000-0003-4388-7783
Ashok Litwin-Kumar ⓘ http://orcid.org/0000-0003-2422-6576
Toshihide Hige ⓘ http://orcid.org/0000-0002-0007-3192
Yoshinori Aso ⓘ http://orcid.org/0000-0002-2939-1688

## Decision letter and Author response

Decision letter https://doi.org/10.7554/eLife.79042.sa1
Author response https://doi.org/10.7554/eLife.79042.sa2

# Additional files

## Supplementary files

• Supplementary file 1. Neurotransmitter prediction and a full connection matrix for MBONs, DANs and 396 interneurons cell types. Numbers in column B-G are numbers of presynaptic sites that are predicted to be designated neurotransmitters. EM id in column K is an identification number in EM hembrain data. The other columns are the connection matrix. Top row indicates the direction of connections. For instance, 153 in the raw 5 of column M indicate the number of connections from MBON01 to SMP108, while 166 in the raw5 of column BD indicate the number of connections from SMP108 to PAM02. For the cell type consisting of multiple cells, a summed number of connections

are shown.

• MDAR checklist

### Data availability

The confocal images of expression patterns are available online (http://www.janelia.org/split-gal4). The source data for each figure are included in the manuscript.

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

# Appendix 1

## Appendix 1—key resources table

| Reagent type (species) or resource | Designation | Source or reference | Identifiers | Additional information |
|---|---|---|---|---|
| strain, strain background (*Drosophila melanogaster*) | Canton S | Martin Heisenberg | N.A. | |
| strain, strain background (*Drosophila melanogaster*) | *20xUAS-CsChrimson-mVenus attP18* | *Klapoetke et al., 2014*; PMID: 24509633 | N.A. | |
| strain, strain background (*Drosophila melanogaster*) | *10XUAS-Chrimson88-tdTomato attP1* | *Klapoetke et al., 2014*; PMID: 24509633 | N.A. | |
| strain, strain background (*Drosophila melanogaster*) | 13XLexAop2-IVS-ChrimsonR-mVenus-p10 attP18 | Vivek Jayaraman | N.A. | |
| strain, strain background (*Drosophila melanogaster*) | 20XUAS-syn21-mScarlet-opt-p10 su(Hw)attp8 | Glenn Turner | N.A. | |
| strain, strain background (*Drosophila melanogaster*) | *pJFRC200-10xUAS-IVS-myr::smGFP-HA in attP18* | *Nern et al., 2015*; PMID: 25964354 | N.A. | |
| strain, strain background (*Drosophila melanogaster*) | *pJFRC225-5xUAS-IVS-myr::smGFP-FLAG in VK00005* | *Nern et al., 2015*; PMID: 25964354 | N.A. | |
| strain, strain background (*Drosophila melanogaster*) | *pBPhsFlp2::PEST in attP3* | *Nern et al., 2015*; PMID: 25964354 | N.A. | |
| strain, strain background (*Drosophila melanogaster*) | *pJFRC201-10XUAS-FRT>STOP > FRT-myr::smGFP-HA in VK0005* | *Nern et al., 2015*; PMID: 25964354 | N.A. | |
| strain, strain background (*Drosophila melanogaster*) | *pJFRC240-10XUAS-FRT>STOP > FRT-myr::smGFP-V5-THS-10XUAS-FRT>STOP > FRT-myr::smGFP-FLAG_in_ su(Hw)attP1* | *Nern et al., 2015*; PMID: 25964354 | N.A. | |
| strain, strain background (*Drosophila melanogaster*) | *LexAop2-DA2m VK00005* | *Sun et al., 2020*; PMID: 33087905 | N.A. | |
| strain, strain background (*Drosophila melanogaster*) | MB043-split-LexA | This paper | N.A. | Available from Aso lab |
| strain, strain background (*Drosophila melanogaster*) | *empty-split-GAL4 (p65ADZp attP40, ZpGAL4DBD attP2)* | *Seeds et al., 2014*; PMID: 25139955 | N.A. | |
| strain, strain background (*Drosophila melanogaster*) | MB032B split-GAL4 | *Aso et al., 2014*; PMID: 25535793 | N.A. | |
| strain, strain background (*Drosophila melanogaster*) | MB043C split-GAL4 | *Aso et al., 2014*; PMID: 25535793 | N.A. | |
| strain, strain background (*Drosophila melanogaster*) | MB109B split-GAL4 | *Aso et al., 2014*; PMID: 25535793 | N.A. | |
| strain, strain background (*Drosophila melanogaster*) | MB213B split-GAL4 | *Aso et al., 2014*; PMID: 25535793 | N.A. | |
| strain, strain background (*Drosophila melanogaster*) | MB315C split-GAL4 | *Aso et al., 2014*; PMID: 25535793 | N.A. | |
| strain, strain background (*Drosophila melanogaster*) | SS33917 split-GAL4 | This paper | N.A. | Available from Aso lab |
| strain, strain background (*Drosophila melanogaster*) | SS45234 split-GAL4 | This paper | N.A. | Available from Aso lab |
| strain, strain background (*Drosophila melanogaster*) | SS67221 split-GAL4 | This paper | N.A. | Available from Aso lab |
| strain, strain background (*Drosophila melanogaster*) | UAS-TeNT | *Keller et al., 2002*; PMID: 11810637 | N.A. | |
| antibody | anti-GFP (rabbit polyclonal) | Invitrogen | A11122 RRID:AB_221569 | 1:1000 |
| antibody | anti-Brp (mouse monoclonal) | *Developmental Studies Hybridoma Bank* | nc82 RRID:AB_2341866 | 1:30 |

*Appendix 1 Continued on next page*

*Appendix 1 Continued*

| Reagent type (species) or resource | Designation | Source or reference | Identifiers | Additional information |
|---|---|---|---|---|
| antibody | anti-ChAT (mouse monoclonal) | *Developmental Studies Hybridoma Bank* | ChAT4B1 RRID:AB_528122 | 1:50 |
| antibody | anti-HA-Tag (mouse monoclonal) | Cell Signaling Technology | C29F4; #3724 RRID:AB_10693385 | 1:300 |
| antibody | anti-FLAG (rat monoclonal) | Novus Biologicals | NBP1-06712 RRID:AB_1625981 | 1:200 |
| antibody | anti-V5-TAG Dylight-549 (mouse monoclonal) | Bio-Rad | MCA2894D549GA RRID:AB_10845946 | 1:500 |
| antibody | anti-mous IgG(H&L) AlexaFluor-568 (goat polyclonal) | Invitrogen | A11031 RRID:AB_144696 | 1:400 |
| antibody | anti-rabbit IgG(H&L) AlexaFluor-488 (goat polyclonal) | Invitrogen | A11034 RRID:AB_2576217 | 1:800 |
| antibody | anti-mouse IgG(H&L) AlexaFluor-488 conjugated (donkey polyclonal) | Jackson Immuno Research Labs | 715-545-151 RRID:AB_2341099 | 1:400 |
| antibody | anti-rabbit IgG(H&L) AlexaFluor-594 (donkey polyclonal) | Jackson Immuno Research Labs | 711-585-152 RRID:AB_2340621 | 1:500 |
| antibody | anti-rat IgG(H&L) AlexaFluor-647 (donkey polyclonal) | Jackson Immuno Research Labs | 712-605-153 RRID:AB_2340694 | 1:300 |
| antibody | anti-Mouse IgG (H&L) ATTO 647 N (goat polyclonal) | ROCKLAND | 610-156-121 RRID:AB_10894200 | 1:100 |
| antibody | anti-rabbit IgG (H+L) Alexa Fluor 568 (goat polyclonal) | Invitrogen | A-11036 RRID:AB_10563566 | 1:1000 |
| chemical compound, drug | 3-Octanol | Sigma-Aldrich | 218405 | |
| chemical compound, drug | 4-Methylcyclohexanol | VWR | AAA16734-AD | |
| chemical compound, drug | Pentyl acetate | Sigma-Aldrich | 109584 | |
| chemical compound, drug | Ethyl lactate | Sigma-Aldrich | W244015 | |
| chemical compound, drug | Paraffin oil | Sigma-Aldrich | 18512 | |
| software, algorithm | ImageJ and Fiji | NIH *Schindelin et al., 2012* | https://imagej.nih.gov/ij/ http://fiji.sc/ | |
| software, algorithm | MATLAB | MathWorks | https://www.mathworks.com/ | |
| software, algorithm | Adobe Illustrator CC | Adobe Systems | https://www.adobe.com/products/illustrator.html | |
| software, algorithm | GraphPad Prism 9 | GraphPad Software | https://www.graphpad.com/scientific-software/prism/ | |
| software, algorithm | Python | Python Software Foundation | https://www.python.org/ | |
| software, algorithm | neuPrint | HHMI Janelia | https://doi.org/10.25378/janelia.12818645.v1 | |
| software, algorithm | Cytoscape | *Shannon et al., 2003* | https://cytoscape.org/ | |
| software, algorithm | NeuTu | *Zhao et al., 2018* | https://github.com/janelia-flyem/NeuTu; *janelia-flyem, 2021* | |
| software, algorithm | ScanImage | Vidrio Technologies | https://vidriotechnologies.com/ | |
| software, algorithm | VVDveiwer | HHMI Janelia | https://github.com/takashi310/VVD_Viewer | |

*Appendix 1 Continued on next page*

*Appendix 1 Continued*

| Reagent type (species) or resource | Designation | Source or reference | Identifiers | Additional information |
|---|---|---|---|---|
| other | Grade 3 MM Chr Blotting Paper | Whatmann | 3030–335 | Used in glass vials with paraffin-oil diluted odours |
| other | mass flow controller | Alicat | MCW-200SCCM-D | Mass flow controller used for the olfactory arena |

