## [Editor Report]

Second-order conditioning is a higher form of learning where a previously conditioned stimulus is used to condition the perception of another stimulus. The authors have used the fly to identify a neural circuit in the insect mushroom body underpinning this fundamental ability of higher animals. This important work elegantly combines neural circuit mapping, electrophysiology, and modelling to put forward a compelling, mechanistic model for this highly conserved form of learning.

---

## [Decision Letter]

**Decision letter after peer review:**

Thank you for submitting your article "Hierarchical architecture of dopaminergic circuits enables second-order conditioning in *Drosophila*" for consideration by *eLife*. Your article has been reviewed by 3 peer reviewers, one of whom is a member of our Board of Reviewing Editors, and the evaluation has been overseen by K VijayRaghavan as the Senior Editor. The reviewers have opted to remain anonymous.

Essential revisions:

As you will see, the reviewers were impressed by the data and are in support of publication. In particular, reviewer 2 has noted several points that should be addressed before a final decision can be reached. Please address all points in a detailed point-by-point response and correct all errors, improve presentation where requested, and, if possible, add additional data to support your claims.

*Reviewer #1 (Recommendations for the authors):*I find the work very interesting and have nothing to suggest in terms of additional experiments. I found a few small typos that would need fixing:for instance line 338: PMA should be PAM

Any idea why alpha1 that's so isolated plays this role? Do you think the isolation is a feature? And what is the role of SMP108 and the other interneurons in the biological sense?

*Reviewer #2 (Recommendations for the authors):*

1. In Ichinose et al. 2015, alpha1 MBONs are proposed to form feedback input to DANs via NMDARs. A key to the circuit model presented here seems to be that the α 1 MBON actually inhibits downstream targets (glutamate can be either inhibitory or excitatory in the fly nervous system). I suggest that the authors perform functional connectomics in the presence of GluCl antagonists to a next candidate neuron in line (Figure 3). Alternatively, they could apply glutamate focally to a downstream interneuron identified to further drive this point home (that is, of course, only if they have appropriate driver lines available). On a different note: will the code for the hemibrain neurotransmitter prediction be available along with this manuscript?

2. Concerning data display in Figure 2: in this reviewer's opinion it is difficult to see the effect on voltage responses from the average/SEM display, I would suggest showing the individual traces. What exactly is quantified in panel C? I was only able to retrieve the onset of quantification after odor delivery from the methods section – are these 'peak values`? I would also recommend showing the mentioned absence of current-induced action potentials. Please also show the expression patterns of the lines used in Figure 2 and S2-1.

3. Figure 1 convincingly shows the role of distinct DANs in first order conditioning. Does MB043c block affect first order memory performance? The authors could also consider an acute block of MB043C neurons during and/or following first order conditioning. If the shibire background does not work, would using GtACR not make sense? The same holds true for the experiments shown in Figure 6. That said, while a temporally higher resolved knowledge of when the output of the neurons is needed would be very interesting, this could be beyond the scope of this manuscript. Minor: the values for the y-axis are missing in Figure 1 G.

---

## [Author Response]

Reviewer #1 (Recommendations for the authors):I find the work very interesting and have nothing to suggest in terms of additional experiments. I found a few small typos that would need fixing:for instance line 338: PMA should be PAM

We corrected the typos.

Any idea why alpha1 that's so isolated plays this role? Do you think the isolation is a feature? And what is the role of SMP108 and the other interneurons in the biological sense?

Anatomical isolation and the slow learning rate of the α1 compartment likely ensure that flies form a second-order memory only when the first-order memory was formed based on experience of intense/repeated reward. We therefore speculate that this is a circuit feature that was crafted during evolution.

In another manuscript in preparation for submission to *eLife*, we will report functions of SMP353/354 (a cluster of presynaptic neurons of SMP108) in driving memory-based anemotaxis (i.e. oriented movement to the current of airflow). For better understanding of reinforcement signals in SMP353/354 and SMP108, it will be informative to examine whether they respond to reward itself and investigate functions of putative inhibitory interneurons such as SMP553 and LHCENT3.

Reviewer #2 (Recommendations for the authors):1. In Ichinose et al. 2015, alpha1 MBONs are proposed to form feedback input to DANs via NMDARs. A key to the circuit model presented here seems to be that the α 1 MBON actually inhibits downstream targets (glutamate can be either inhibitory or excitatory in the fly nervous system).

We cited Ichinose et al., and other papers that proposed excitatory connection from glutamatergic MBONs to downstream neurons.

“Glutamate functions as an inhibitory neurotransmitter with glutamate-gated-chloride channel (Liu and Wilson, 2013), although activity of glutamatergic MBONs can have a net excitatory effect on DANs (Cohn et al., 2015; Ichinose et al., 2015; Zhao et al., 2018).”

I suggest that the authors perform functional connectomics in the presence of GluCl antagonists to a next candidate neuron in line (Figure 3). Alternatively, they could apply glutamate focally to a downstream interneuron identified to further drive this point home (that is, of course, only if they have appropriate driver lines available).

We are preparing another manuscript to *eLife* about the function of the immediate downstream neurons of MBON-α1 (i.e. SMP353/354 in Figure 4) in memory-guided anemotaxis. In that manuscript, we will report electrophysiological data that SMP353/354 were inhibited by optogenetic activation of MBON-α1.

**Author response image 1. sa2fig1:** Functional connectivity between MBON-α1 and UpWiNs. (A) Functional connectivity between MBON-α3 and UpWINs. Chrimson88-tdTomato was expressed in MBON-α3 by MB082C split-GAL4, and the photostimulation responses were measured by whole-cell current-clamp recording in randomly selected SMP353/354 labeled by R64A11-LexA. 2 out of 6 neurons (4 flies) showed excitatory response. Mean voltage traces from individual connected (orange) and unconnected neurons (gray) are overlaid. Connection was strong enough to elicit spikes (black; single-trial response in one of the connected neurons). Magenta vertical line indicates photostimulation (10 msec). (B) Functional connectivity between MBON-α1 and SMP353/354. Chrimson88-tdTomato expression in MBON-α1 was driven by MB310C split-GAL4. 4 out of 17 neurons (12 flies) showed inhibitory response. Mean voltage traces from individual connected (green) and unconnected neurons (gray) are overlaid. Figure 4A and B of Aso et al 2022 available at https://doi.org/10.1101/2022.12.21.521497.

On a different note: will the code for the hemibrain neurotransmitter prediction be available along with this manuscript?

The preprint and github description is available for the neurotransmitter predictions. Funke lab is actively working on easier access to the neurotransmitter predictions.

https://www.biorxiv.org/content/10.1101/2020.06.12.148775v1 https://github.com/funkelab/synister

2. Concerning data display in Figure 2: in this reviewer's opinion it is difficult to see the effect on voltage responses from the average/SEM display, I would suggest showing the individual traces. What exactly is quantified in panel C? I was only able to retrieve the onset of quantification after odor delivery from the methods section – are these 'peak values`? I would also recommend showing the mentioned absence of current-induced action potentials.

In (original) Figure 2C, the response magnitude was calculated as the depolarization during the entire response period from the onset to 0.6 s after the offset of the 20-s odor presentation. We did so because we observed the main effect of the conditioning in the sustained odor response rather than the peak response right after the odor onset. In fact, there was no significant change in the peak response. The quantification method was described in the methods section in the original manuscript, but we realized that it was not clear enough. To make it more accessible to readers, we added this information to the figure legend together with the statistics of the peak amplitude. Showing individual traces will not help because the vertical spread of the voltage traces in panels B and D reflects actual temporal fluctuations of the mean voltage, not SEM; there are more fluctuations in individual traces. We also believe that the main effect (i.e. depression) in the sustained odor response is clearly visible in our original plots. We nonetheless made the lines thinner to help visualization.

We added the data to show the responses to current injection in MBON-γ5β′2a as Figure 3—figure supplement 3.

Please also show the expression patterns of the lines used in Figure 2 and S2-1.

The expression patterns of the lines are shown in new Figure 3—figure supplement 1.

3. Figure 1 convincingly shows the role of distinct DANs in first order conditioning. Does MB043c block affect first order memory performance?

We added Figure 2D (left). MB043C>TNT flies showed an impaired preference to the S1+ odor one day after training. This is consistent with Ichinose et al., 2015 *eLife*, which showed that blocking of PAM-α1 DANs with temperature-sensitive Shibire during odor-sugar training impaired the memory tested 1 day after the training. In that paper, blocking PAM-α1 DANs during training and test did not impair immediate memory. Therefore, the impairment of second-order conditioning by MB043C>TNT flies is likely due to the lack of first-order memory in the α1, although we cannot fully exclude the possibility that PAM-α1 is required to form second-order memory in the α1 compartment.

The authors could also consider an acute block of MB043C neurons during and/or following first order conditioning. If the shibire background does not work, would using GtACR not make sense? The same holds true for the experiments shown in Figure 6. That said, while a temporally higher resolved knowledge of when the output of the neurons is needed would be very interesting, this could be beyond the scope of this manuscript.

The problem with Shibire experiment was not the genetic background of the Shibire but the high temperature. Wild type flies failed to form second-order memory at restrictive temperatures, presumably because of premature extinction of the first-order memory and/or aversiveness of heat. We planned GtACR1 experiments but wild type flies also failed to perform odor-to-odor second-order conditioning in the presence of intense green light. RubyACR, a red-shifted optogenetic inhibitor (Govorunova et al., 2020 PNAS; https://doi.org/10.1073/pnas.200598111) is a promising alternative, but it will take time to make UAS lines for it. We very much appreciate the reviewer's agreement that temporal requirements of DANs can be the scope of a future study.